# Responses to membrane potential-modulating ionic solutions measured by magnetic resonance imaging of cultured cells and in vivo rat cortex

Kyeongseon Min[1], Sungkwon Chung[2], Seung-Kyun Lee[3,4], Jongho Lee[1], Phan Tan Toi[3,4], Daehong Kim[5], Jung Seung Lee[3,4], Jang-Yeon Park[3,4]*

[1]Department of Electrical and Computer Engineering, Seoul National University, Seoul, Republic of Korea; [2]Department of Physiology, Sungkyunkwan University School of Medicine, Suwon, Republic of Korea; [3]Department of Biomedical Engineering, Sungkyunkwan University, Seoul, Republic of Korea; [4]Department of Intelligent Precision Healthcare Convergence, Sungkyunkwan University, Seoul, Republic of Korea; [5]National Cancer Center, Goyang-si, Republic of Korea

*For correspondence:
jyparu@skku.edu

Competing interest: The authors declare that no competing interests exist.

## eLife Assessment

The authors show MRI relaxation time changes that are claimed to originate from cell membrane potential changes. This would be a substantial contribution if true because it may provide a mechanism whereby membrane potential changes could be inferred noninvasively. However, the membrane potential manipulations applied here are performed on a slow time scale and are known to induce cell swelling. Cell swelling has been previously shown to affect relaxation time. Experiments could be performed to rule out this hypothesis, but the authors have chosen not to perform these experiments. The study is therefore **useful**, but the evidence is **incomplete**.

**Abstract** Membrane potential plays a crucial role in various cellular functions. However, existing techniques for measuring membrane potential are often invasive or have limited recording depth. In contrast, MRI offers noninvasive imaging with desirable spatial resolution over large areas. This study investigates the feasibility of utilizing MRI to detect responses of cultured cells and in vivo rat cortex to membrane potential-modulating ionic solutions by measuring magnetic resonance parameters. Our findings reveal that depolarizing (or hyperpolarizing) ionic solutions increase (or decrease) the $T_2$ relaxation time, while the ratio of bound to free water protons shows the opposite trend. These findings also suggest a potential approach to noninvasively detect changes in membrane potential using MRI.

## Introduction

Membrane potential is a fundamental property of all living cells, influencing crucial cell functions (**Abdul Kadir et al., 2018**) such as neuronal and myocyte excitability, volume control, cell proliferation, and secretion. In the fields of neuroscience, membrane potential is of significant importance, as neural activities arise from the dynamic propagation of this electric potential. From a clinical perspective, deviations from normal membrane potential levels contribute to various diseases, including seizures (**Jefferys, 1995**), arrhythmia (**Helfant, 1986**), and hypoglycemia (**Kane et al., 1996**). Given

its scientific and clinical importance, effective methods to detect changes in membrane potential have long been sought.

The intracellular recording technique using sharp glass electrodes is a commonly used method to detect changes in membrane potential (*Ling and Gerard, 1949*; *Neher and Sakmann, 1976*). It provides real-time absolute recordings of membrane potential. Optical imaging is another major approach to detect changes in membrane potential. Voltage-sensitive dyes enable voltage imaging at the cellular level (*Tasaki et al., 1968*). Fluorescent calcium indicators detect calcium transients associated with neuronal activation (*Tsien, 1980*). Label-free optical imaging techniques (*Zhou et al., 2021*) utilize various contrast mechanisms such as cell membrane deformation, which are directly coupled with changes in membrane potential. Despite their efficacy, these methods have limitations when applied to intact biological systems due to their invasive nature, requiring procedures such as craniotomy.

As noninvasive techniques for directly or indirectly detecting brain activation in vivo, several imaging modalities have been developed, including EEG (*Berger, 1929*), MEG (*Cohen, 1968*), and MRI (*Mansfield, 1977*). EEG and MEG detect electric potentials on the scalp and extracranial magnetic fields, respectively, which are directly induced by neuronal activity in the brain. Although these techniques are noninvasive and provide excellent temporal resolution of milliseconds or less, they are constrained by shallow imaging depth and spatial localization challenges (*He et al., 2018*).

In contrast, MRI enables noninvasive imaging with good spatial resolution of millimeters over a large brain volume, making it an appropriate tool for in vivo functional brain imaging. To date, the mainstream of functional MRI (fMRI) utilizes hemodynamic responses driven by brain activation, such as the blood oxygen level-dependent (BOLD) effect (*Ogawa et al., 1990*). However, while the BOLD contrast mechanism reflects dynamic changes in neuronal activity through neurovascular coupling, it provides inherently indirect and relatively slow, hemodynamic-responsive information of brain function (*Logothetis et al., 2001*). On the other hand, many studies have attempted to explore the possibility of using MRI to directly detect neuronal activity (*Bandettini et al., 2005*; *Roth, 2023*). These studies utilized neuronal current-dependent signal phase shifts (*Petridou et al., 2006*; *Wijesinghe and Roth, 2009*; *Sundaram et al., 2016*) and magnitude decay (*Kamei et al., 1999*; *Xiong et al., 2003*; *Chow et al., 2006*; *Truong et al., 2019*), the Lorentz effect (*Truong and Song, 2006*), high temporal resolution (*Sundaram et al., 2010*; *Toi et al., 2022*), ghost artifacts (*Paley et al., 2009*), or cell swelling (*Le Bihan et al., 2006*; *Bai et al., 2016*), while concerns about sensitivity exist (*Chu et al., 2004*; *Konn et al., 2004*; *Parkes et al., 2007*; *Miller et al., 2007*; *Roth and Basser, 2009*; *Luo et al., 2009*).

In this study, we investigated the possibility of using MRI to detect membrane potential changes induced by modulating ionic solutions. Specifically, we explored how $T_2$ relaxation time and magnetization transfer (MT) correlate with membrane potential changes, both in vitro and in vivo. In vitro experiments utilized two homogeneous and electrically non-active cells, that is neuroblastoma (SH-SY5Y) and leukemia (Jurkat) cell lines, providing a controlled environment free from hemodynamic effects. This aided in accurately evaluating changes in MR parameters with membrane potential. In vivo experiments, conducted on a rat model with a craniotomy-exposed cortex, aimed to reproduce the in vitro findings.

## Results

### In vitro changes in MR parameters induced by membrane potential

A non-excitable neuroblastoma cell line, SH-SY5Y, was selected to investigate the relationship between MR parameters and membrane potential modulated by ionic solutions. After culturing SH-SY5Y cells, they were suspended in extracellular media with various potassium ion concentrations ([K+]), while maintaining constant osmolarity by adjusting [Na+]. As the control condition, [K+]=4.2 mM was selected. For depolarization conditions, [K+]=20, 40, and 80 mM were used. For hyperpolarization conditions, [K+]=0.2 and 1 mM were used. The suspended cells were concentrated with centrifugation in an acrylic container and scanned in a 9.4T preclinical MRI system. The imaging slice was positioned 0.5 mm below the pellet-extracellular media interface to ensure that signals were acquired from the cell pellet. Under each condition, $T_2$ and MT parameters such as pool size ratio (PSR) and magnetization transfer rate ($k_{mf}$) were measured (*Figure 1*). The PSR value represents the ratio of hydrogen protons in macromolecules and free water, and $k_{mf}$ represents the magnetization transfer

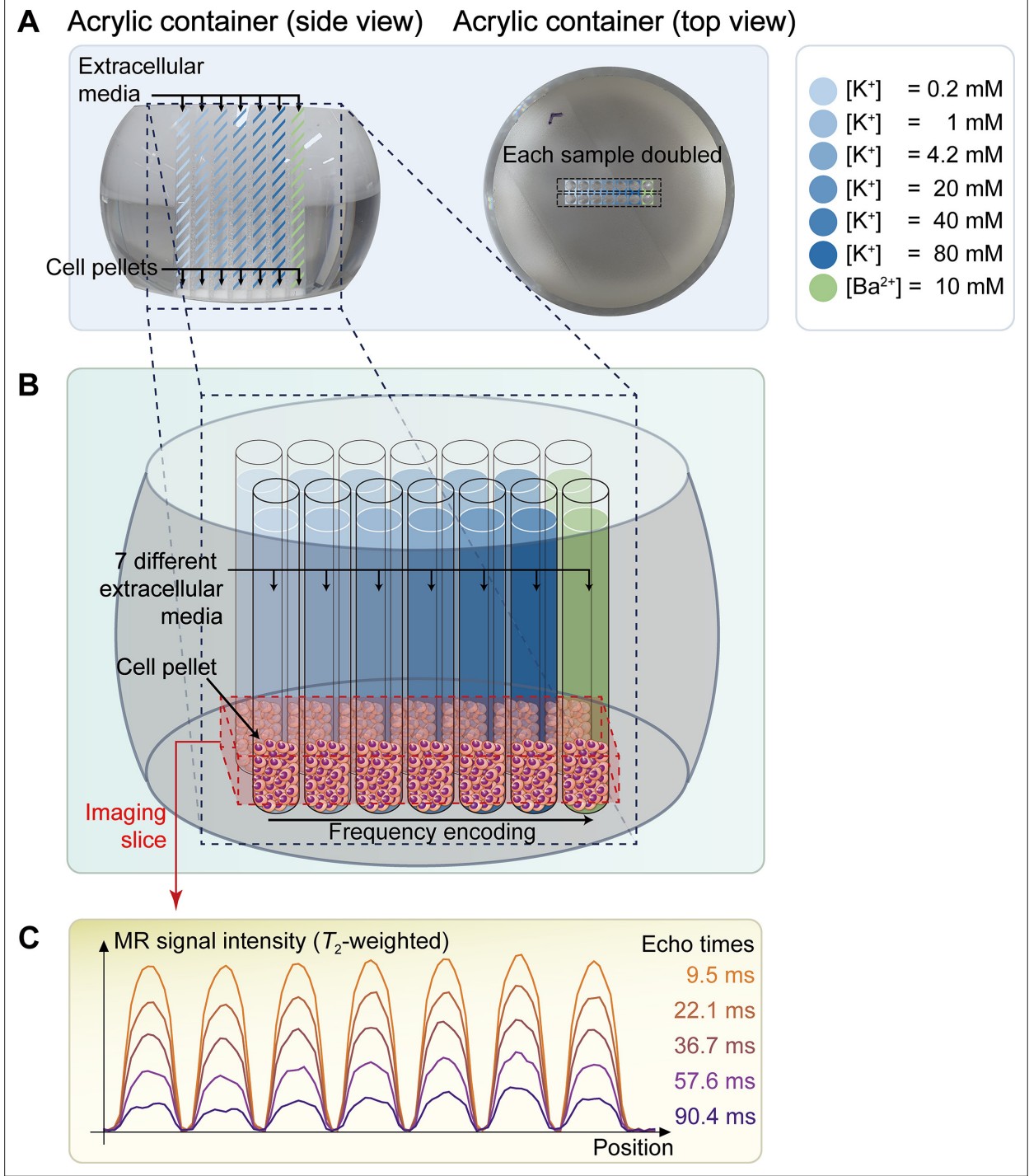

**Figure 1.** The schematic diagram of the in vitro experiment. (**A**) The picture on the left displays a side view of a double-sided cut spherical acrylic container with fabricated wells filled with extracellular media and cell pellets. As depicted in the top-view picture on the right, fourteen wells (matrix = 2 × 7) were created on the acrylic container, allowing each of the seven samples with six different $K^+$ concentrations ($[K^+]$=0.2–80 mM) and one $Ba^{2+}$ concentration ($[Ba^{2+}]$=10 mM) to be doubled in the same column for improved signal-to-noise ratio (SNR) in MR signal acquisition. (**B**) The image illustrates the configuration after loading cells into the wells and pelleting them at the bottom of the wells. The imaging slice was positioned 0.5 mm below the pellet-media interface to acquire signals predominantly from the cell pellets. (**C**) Representative one-dimensional $T_2$-weighted MR signals with 5 selected echo times out of a total of 50 acquired echo times.

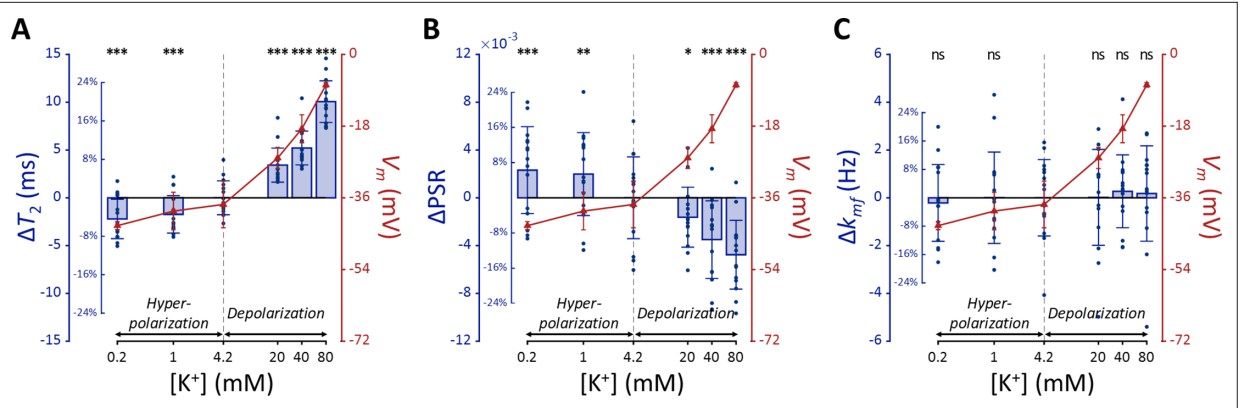

**Figure 2.** MR parameters and membrane potential ($V_m$) of SH-SY5Y cells versus extracellular K$^+$ concentrations ([K$^+$]). Changes in (**A**) $T_2$, (**B**) PSR, and (**C**) $k_{mf}$ are displayed with blue bars (n=15). Membrane potentials are plotted with red triangles (n=3). The abscissa is in logarithmic scale. Error bars denote standard deviation. Statistical significance of changes in MR parameters is marked with asterisks (ns: p>0.05, *: p<0.05, **: p<0.01, ***: p<0.001).

rate of hydrogen protons from macromolecules to free water. In addition, the membrane potential of SH-SY5Y cells in each condition was separately measured via patch clamp recording.

The changes of $T_2$, PSR, and $k_{mf}$ in SH-SY5Y cells when the membrane potential ($V_m$) was modulated by varying [K$^+$] are shown in **Figure 2**, alongside the actual $V_m$ measured via patch clamp recordings. We conducted statistical analyses to assess the effect of changes in $V_m$ from the control condition ($\Delta V_m$) on the MR parameters. A linear mixed-effect model was applied to account for inter-sample variability and repeated measurements. This model included MR parameters as dependent variables, $\Delta V_m$ as a fixed effect, and cell batch as a random effect. The analysis yielded the following relationships:

$$T_2 \text{ (ms)} = 49.7 \left(1 + 0.00687\,\Delta V_m\right) \tag{1}$$

$$\text{PSR} = 0.0377 \left(1 - 0.00542\,\Delta V_m\right) \tag{2}$$

$$k_{\text{mf}} \text{ (Hz)} = 14.8 \left(1 + 0.000648\,\Delta V_m\right) \tag{3}$$

The effects of $\Delta V_m$ on $T_2$ and PSR were both significant (p<0.0001), indicating an increase in $T_2$ and a decrease in PSR during depolarization at high [K$^+$], with the opposite trend during hyperpolarization at low [K$^+$]. On the other hand, the effect of $\Delta V_m$ on $k_{mf}$ was not significant (P=0.360).

Subsequent post-hoc analyses compared each experimental condition to the control using Dunnett's test to account for multiple comparisons (**Figure 2**). During depolarization induced by the highest [K$^+$] (80 mM, $\Delta V_m$ = 30.0 mV), changes in MR parameters were observed as a 20.0% increase in $T_2$ ($\Delta T_2$=10.1ms, p<0.0001) and a 12.9% decrease in PSR ($\Delta$PSR = −0.00476, p<0.0001). Conversely, during hyperpolarization induced by the lowest [K$^+$] (0.2 mM, $\Delta V_m$ = −5.33 mV), $T_2$ decreased by 4.40% ($\Delta T_2$ = −2.21ms, p<0.0001) and PSR increased by 6.28% ($\Delta$PSR = 0.00231, p<0.0005). However, changes in $k_{mf}$ were not significant across all conditions (p>0.05). These findings from in vitro SH-SY5Y cell experiments suggest that MR parameters, such as $T_2$ and PSR, exhibit sufficient sensitivity to detect alterations in membrane potential induced by varying [K$^+$], including both depolarization and hyperpolarization.

## Using a K$^+$ channel blocker

In this experiment, we investigated whether depolarization induced by altering potassium permeability with barium ions (Ba$^{2+}$) would affect MR parameters similarly to depolarization induced by varying [K$^+$], thereby further validating our findings. For this purpose, we administered barium ions (Ba$^{2+}$) at a concentration of 10 mM to induce depolarization while maintaining constant osmolarity by adjusting [Na$^+$]. Ba$^{2+}$ was chosen because it inhibits several types of two-pore-domain potassium channels (**Lesage and Lazdunski, 2000**; **Ma et al., 2011**), which predominantly regulate the resting membrane potential. Previous studies have confirmed that Ba$^{2+}$ depolarizes the membrane potential in SH-SY5Y and Jurkat cells (**Vaughan et al., 1995**; **Pottosin et al., 2008**).

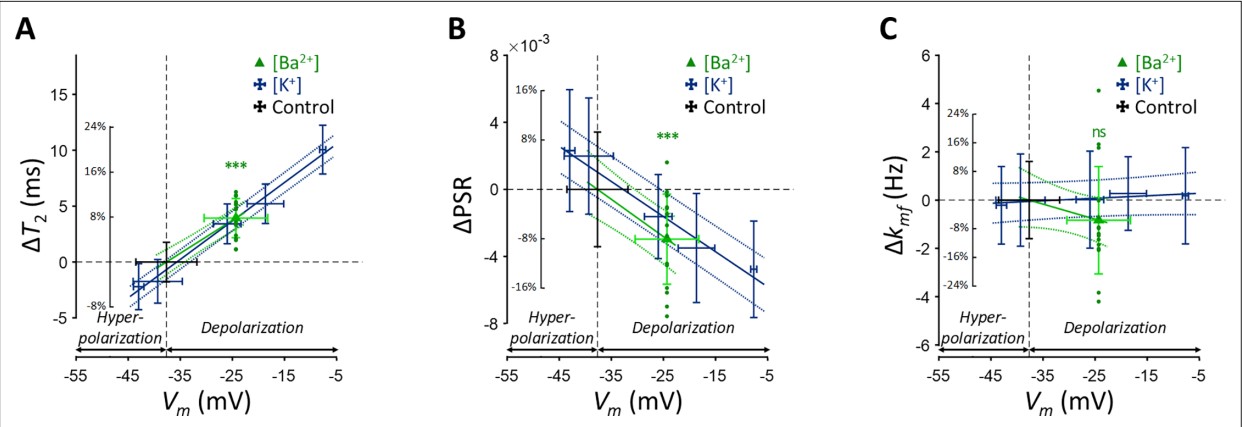

**Figure 3.** Changes in (**A**) $T_2$, (**B**) PSR, and (**C**) $k_{mf}$ of SH-SY5Y cells across experimental conditions: [K⁺]=0.2–80 mM (blue cross) and [Ba²⁺]=10 mM (green triangle), compared to the control condition (black cross). Data from fifteen experiments (n=15) are displayed. Linear regression lines for [K⁺] data (blue solid line) and [Ba²⁺] data (green solid line) are drawn along with dotted lines representing 95% confidence intervals. Error bars denote standard deviation. Statistical significance of changes in MR parameters with [Ba²⁺]=10 mM is marked with asterisks (ns: p>0.05, *: p<0.05, **: p<0.01, ***: p<0.001).

The Ba²⁺-induced depolarization condition was compared with K⁺-induced depolarization and hyperpolarization conditions (**Figure 3**). To compare the effects of [Ba²⁺] and [K⁺] on MR parameters, a linear mixed-effect model was utilized. This model included MR parameters as dependent variables, $\Delta V_m$ and its interaction with a group variable (indicating whether $\Delta V_m$ was induced by [Ba²⁺] or [K⁺]) as fixed effects, and cell batch as a random effect. This analysis revealed no significant interactions for all MR parameters assessed (p=0.182 for $T_2$, p=0.788 for PSR, and p=0.0890 for $k_{mf}$). These findings suggest that changes in $T_2$ and PSR by membrane potential do not depend on the specific method of altering the membrane potential, whether by varying [K⁺] or applying [Ba²⁺]. This implies that if the changes in MR parameters observed with varying [K⁺] were not primarily due to membrane potential but were due to a unique K⁺-related mechanism, then experiments using [Ba²⁺] would not result in similar changes.

Subsequent post-hoc analyses compared the [Ba²⁺]-induced depolarization to the control using Dunnett's test to account for multiple comparisons (**Figure 3**). In response to depolarization caused by [Ba²⁺]=10 mM, $T_2$ increased by 7.82% ($\Delta T_2$=3.93ms, p<0.0001) and PSR decreased by 8.06% ($\Delta$PSR = −0.00297, p<0.0001). The change in $k_{mf}$ was not significant ($\Delta k_{mf}$ = −0.832 Hz, p=0.263). The depolarization of membrane potential induced by [Ba²⁺] was measured as $\Delta V_m$ = 13.3 mV by patch clamp recording.

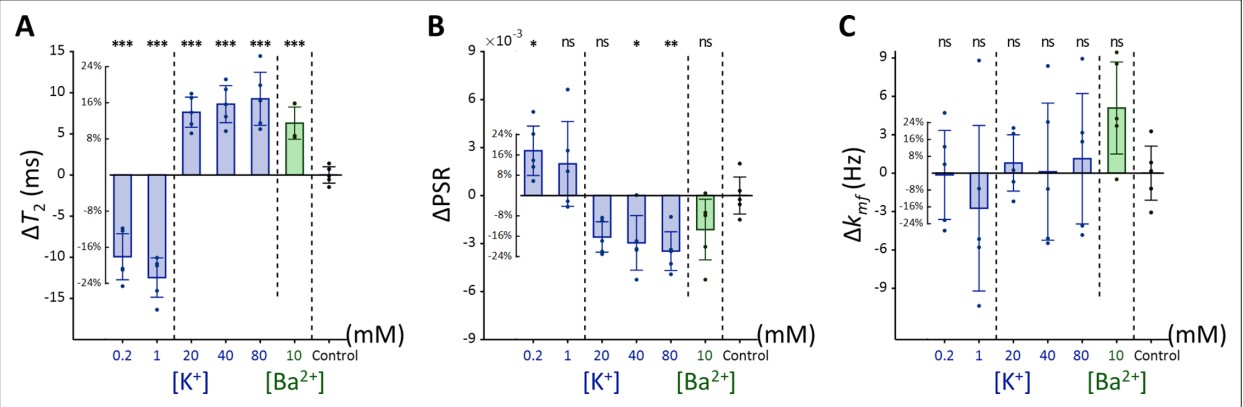

**Figure 4.** Changes in (**A**) $T_2$, (**B**) PSR, and (**C**) $k_{mf}$ of Jurkat cells across experimental conditions: [K⁺]=0.2–80 mM (blue bar) and [Ba²⁺]=10 mM (green bar), compared to the control condition of [K⁺]=4.2 mM (n=5). Error bars denote standard deviation. Statistical significance of changes in MR parameters is marked with asterisks (ns: p>0.05, *: p<0.05, **: p<0.01, ***: p<0.001).

## Using another cell type

To investigate whether membrane potential-modulating ionic solutions produce similar changes in MR parameters across different cell types, we assessed another cell line, Jurkat, under the same experimental conditions applied to SH-SY5Y cells. These conditions included a control condition ([K$^+$]=4.2 mM), hyperpolarization under decreased [K$^+$] conditions ([K$^+$]=0.2 and 1 mM), depolarization under increased [K$^+$] conditions ([K$^+$]=20, 40, and 80 mM), and a Ba$^{2+}$-induced depolarization condition ([Ba$^{2+}$]=10 mM), all maintaining consistent osmolarity by adjusting [Na$^+$].

Each experimental condition was compared to the control using Dunnett's test to account for multiple comparisons (*Figure 4*). As observed with SH-SY5Y cells, Jurkat cells showed significant positive changes in $T_2$ and negative changes in PSR under increased [K$^+$] conditions. For example, at [K$^+$]=80 mM, $T_2$ increased by 16.9% ($\Delta T_2$=9.26ms, p<0.0001), PSR decreased by 21.9% ($\Delta$PSR = −0.00347, p<0.01). In contrast, during hyperpolarization at the lowest [K$^+$]=0.2 mM, $T_2$ decreased by 18.1% ($\Delta T_2$ = −9.93ms, p<0.0001), whereas PSR increased by 17.6% ($\Delta$PSR = 0.00280, p<0.05). The depolarization induced by [Ba$^{2+}$]=10 mM resulted in a similar pattern, with $T_2$ increasing by 11.5% ($\Delta T_2$=6.3ms, p<0.0005), although the decrease in PSR was not significant ($\Delta$PSR = −0.00212, p=0.211). Changes in $k_{mf}$ remained non-significant across all conditions (p>0.05). In summary, these findings indicate that detecting membrane potential changes induced by ionic solutions using MR parameters such as $T_2$ and PSR is not specific to a single cell type, although the magnitude of these changes may differ between cell types.

## In vivo changes in $T_2$ by membrane potential

The relationship of $T_2$ values and membrane potential modulated by [K$^+$], observed in the aforementioned SH-SY5Y and Jurkat cell studies, was further explored in an in vivo rat model to validate these findings under physiological conditions. As depicted in *Figure 5*, a craniotomy was performed to expose a 3-mm-diameter region of the rat cerebral cortex, followed by perfusion with artificial cerebrospinal fluid (aCSF) to modulate the membrane potential. Hemodynamic responses were pharmacologically suppressed. MRI scans were performed using a 7T preclinical MRI system to measure $T_2$ in the exposed cortical area. A total of seven rats were used in the experiment that involved modulation of [K$^+$]. The experimental protocol included sequential application of four conditions, each lasting 12 min: a baseline condition at [K$^+$]=3 mM, a depolarization condition at [K$^+$]=40 mM, further depolarization at [K$^+$]=80 mM, followed by a recovery condition using baseline aCSF. The recovery condition was applied to two of the seven rats. To distinguish the effect of aCSF perfusion on $T_2$ from the effect of changes in membrane potential, a control experiment was also conducted using only baseline aCSF for the entire duration (48 min) with another group of five rats.

In *Figure 6A*, a representative $T_2$ map is displayed with an enlarged image defining the ROI beneath the perfusion chamber (width = 1.8 mm, depth = 0.6 mm). The average $T_2$ value within this ROI was estimated from the spatially averaged multi-echo spin-echo signal. A detailed analysis of the quality of these $T_2$ maps is presented in Appendix 3. Changes in $T_2$ value ($\Delta T_2$) were statistically analyzed using a linear mixed-effect model to account for inter-sample variability and repeated measurements. This model included $\Delta T_2$ as a dependent variable, elapsed time and its interaction with the experiment type (i.e. [K$^+$]-modulation or control) as fixed effects, and a random effect for inter-sample variability.

*Figure 6B* shows the results of the statistical analysis. $\Delta T_2$ values were plotted against elapsed time for [K$^+$]-modulation and control experiments. After 12 mins from the initial condition, $T_2$ increased by 1.46% ($\Delta T_2$=0.684ms) in the [K$^+$]-modulation experiment ([K$^+$]=40 mM) and by 0.223% ($\Delta T_2$=0.104ms) in the control experiment ([K$^+$]=3 mM); the $\Delta T_2$ difference (=0.580ms) between these experiments was not statistically significant (p=0.0711). After 24 min, $T_2$ increased by 2.34% ($\Delta T_2$=1.10ms) in the [K$^+$]-modulation experiment ([K$^+$]=80 mM) and by 0.386% ($\Delta T_2$=0.181ms) in the control experiment ([K$^+$]=3 mM), with a significant difference in $\Delta T_2$ (=0.918ms) between them (p=0.00172). Due to the limited sample size in the recovery phase (n=2 out of 7 for [K$^+$]-modulation), comparisons between experiments were not conducted for this recovery phase.

Our observations indicate that in vivo manipulation of membrane potential results in similar trends of $T_2$ changes as those observed in vitro, albeit with a smaller magnitude in the rat cortex. This discrepancy may be attributed to several factors: [K$^+$]-modulation affecting only a small portion of cells within the ROI; limited diffusion of aCSF through the leptomeninges; removal of excessive K$^+$ through clearing mechanisms; and differences in cell types (see the Discussion for more details).

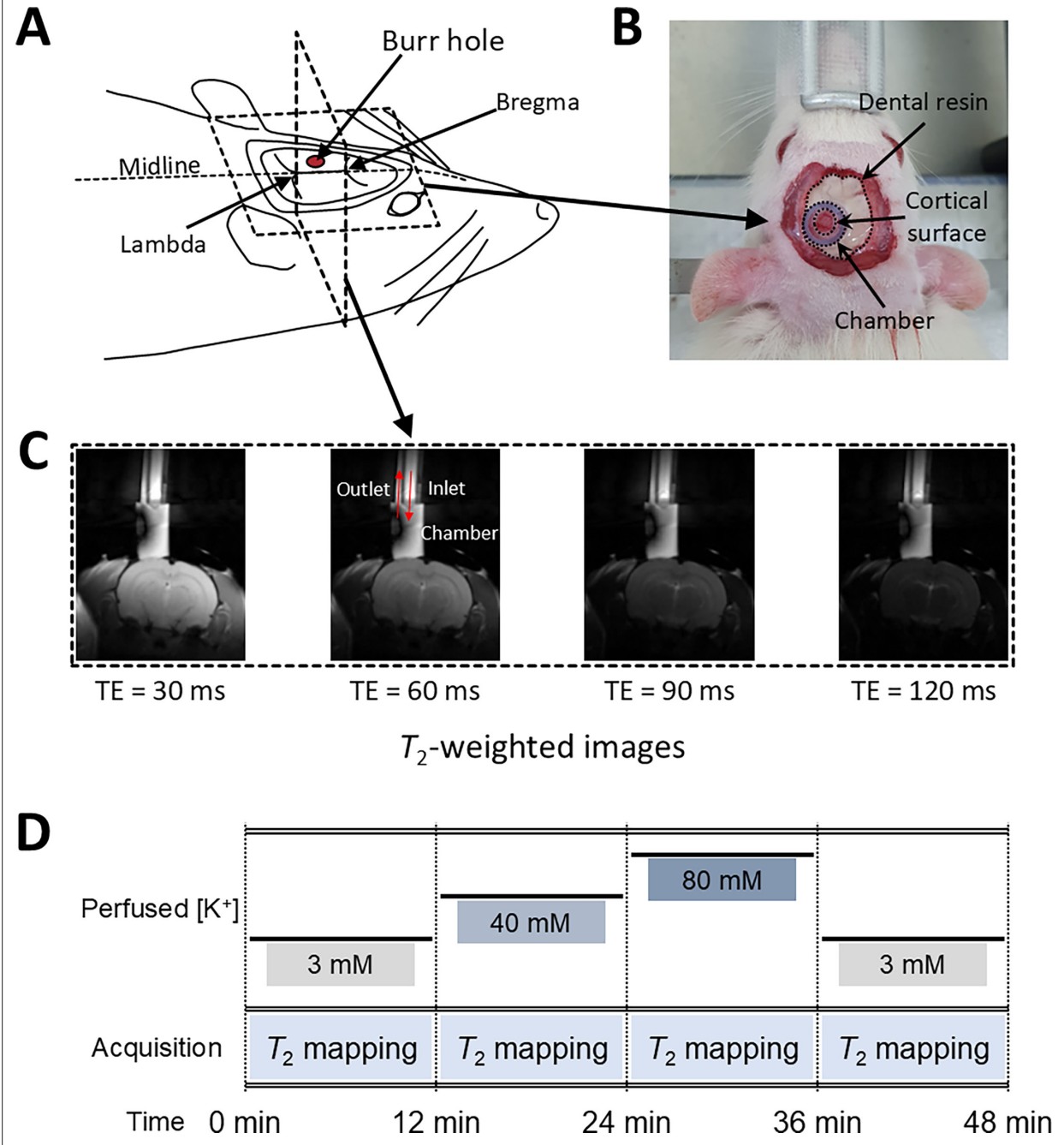

**Figure 5.** Experimental setup for in vivo manipulation of membrane potential. (**A**) A schematic diagram of the rat head post-craniotomy, showing the burr hole centered at 2.5 mm anterior and 2.0 mm lateral to the lambda. (**B**) Photograph of the rat head with a cylindrical chamber fixed over the burr hole, filled with artificial cerebrospinal fluid. (**C**) A representative series of $T_2$-weighted MR images for $T_2$ mapping. The chamber was connected to inlet and outlet perfusion tubes. (**D**) The experimental paradigm of the in vivo rat MR imaging. Four conditions were sequentially applied: control, depolarization, further depolarization, and recovery. Each condition lasted 12 min during which $T_2$ mapping was conducted.

## Discussion

In this study, we demonstrated that MR parameters, specifically $T_2$ relaxation time and pool size ratio (PSR), can detect responses to membrane potential changes modulated by ionic solutions. Our in vitro experiments with cultured cells were designed to exclude physiological factors such as hemodynamic responses and respiration. We observed that depolarization increases $T_2$ and decreases PSR, while hyperpolarization has the opposite effect. In vivo, we pharmacologically suppressed hemodynamic

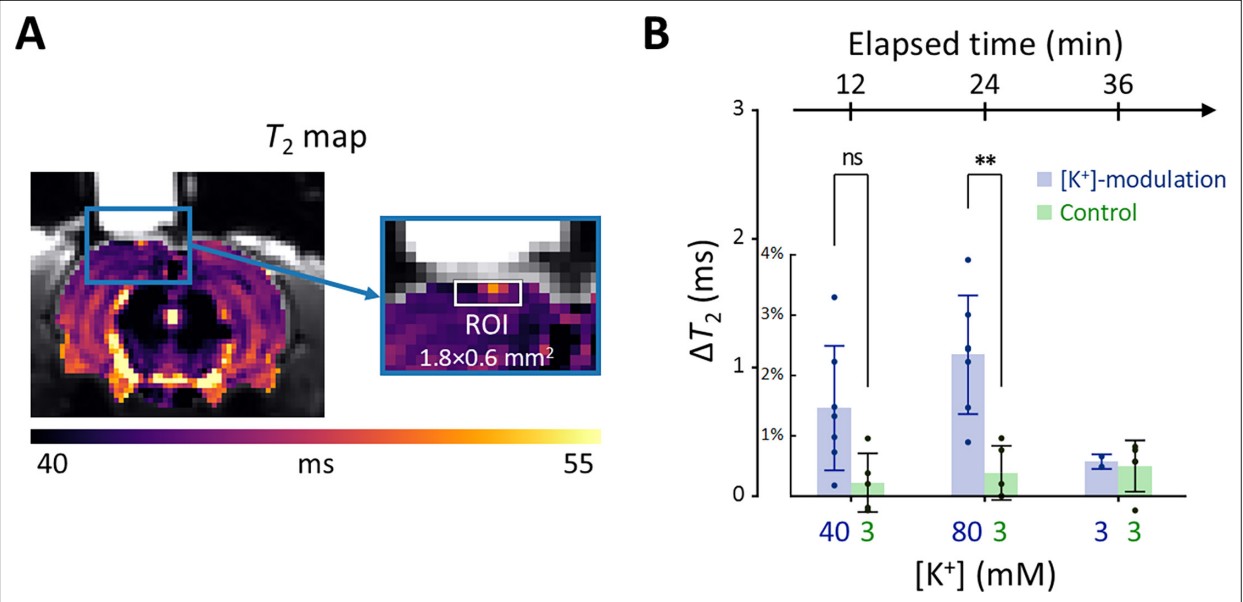

**Figure 6.** Results of the in vivo experiment results in rat models. (**A**) A representative T2 map from a single rat with an enlarged image depicting the ROI for estimating average T2 in the exposed cortical area, marked by a white rectangle (width = 1.8 mm, depth = 0.6 mm). (**B**) The changes in T2 values within the ROI is plotted against elapsed time from the initial conditions. [K$^+$] in the perfused artificial cerebrospinal fluid are indicated on the bottom abscissa. Results from the [K$^+$]-modulation experiments are shown with blue bars, and those from the control experiments are shown with green bars. Statistical significance of the T2 changes is marked with asterisks (ns: $p > 0.05$, *: $p < 0.05$, **: $p < 0.01$).

responses to minimize their impact on $T_2$ measurements. The trend of $T_2$ dependence on membrane potential in vivo was consistent with in vitro findings. However, the magnitude of $T_2$ changes in rat cortex was approximately one-ninth of that observed in SH-SY5Y cells in vitro at [K$^+$]=80 mM.

Other studies have also reported MR-detectable changes in response to extracellular [K$^+$] modulation. For instance, research on spreading depression, which was induced by significantly high [K$^+$] (~1 M) applied topically to rat cortex, has revealed detectable changes in $T_1$, $T_2$, and magnetization transfer ratio changes (*Stanisz et al., 2002*) and spin-lock fMRI signals (*Autio et al., 2014*). Similarly, increased [K$^+$] has been studied in vitro with brain slices, linking cell volume changes to proton density-weighted MR signal changes (*Stroman et al., 2008*). $T_2$ mapping in Jurkat cells with increased [K$^+$] conditions has also been investigated (*Toi et al., 2022*) and linked to cell volume change (*Phi Van et al., 2024*). Our study complements these findings by employing direct membrane potential measurement via patch clamp, testing additional ionic agents such as Ba$^{2+}$, and demonstrating the phenomenon in vivo. Further reinforcement of these findings could be achieved by simultaneous recording of cell volume and other cellular characteristics to elucidate the underlying mechanisms more completely.

Interestingly, the MR responses in Jurkat cells differed from those in SH-SY5Y cells. While SH-SY5Y cells showed a near-linear dependence of MR parameters on log [K$^+$], Jurkat cells displayed more step-like behavior. The reasons for this discrepancy remain unclear, but it suggests that the relationship between membrane potential and MR parameters may not strictly follow a simple log [K$^+$] relationship.

Several factors may contribute to discrepancies between in vivo and in vitro results. For instance, the actual extracellular [K$^+$] experienced by cells in the rat cortex may be lower than that of the perfused aCSF due to diffusion-limiting barriers such as the leptomeninges (*Bradbury et al., 1972*; *Filippidis et al., 2012*), even after the removal of the dura mater. Additionally, removal of excessive K$^+$ by clearing mechanisms (*Walz, 2000*; *O'Donnell, 2009*) may further reduce the [K$^+$] experienced by the cells. Partial volume effects and cell type differences may also contribute to the discrepancy.

From a biophysical perspective, the sensitivity of $T_2$ and PSR to membrane potential likely arises from alterations in cell volume, hydration water, and bulk water. Depolarization or hyperpolarization can lead to cell swelling or shrinking (*Lang et al., 1998*; *Fraser and Huang, 2004*; *Hoffmann et al., 2009*), influencing the proportion of cellular contents within the imaging voxel and thereby affecting

MR parameters. Notably, neuronal or glial cell swelling has been proposed as a possible mechanism underlying diffusion fMRI (*Le Bihan et al., 2006*; *Le Bihan, 2012*; *Mangia et al., 2012*) and previous MRI studies (*Stroman et al., 2008*; *Phi Van et al., 2024*). Although not as sensitive as $T_2$, our $T_1$ measurements (*Appendix 2—figures 1–3*) also exhibited similar trends in response to membrane potential changes. Moreover, hydration water, which has a significantly shorter $T_2$ than bulk water due to slower re-orientational and diffusive motions (*Mathur-De Vré, 1980*), may contribute to the observed MR changes. In particular, the correlation between PSR and membrane potential indicates that depolarization decreases the density of hydration water on the cell membrane within a voxel, thereby reducing PSR and simultaneously increasing $T_2$ due to a corresponding increase in free water, as well as cell swelling. Conversely, hyperpolarization may increase hydration water density, elevating PSR and lowering $T_2$. These interpretations are supported by recent optical studies showing reduced membrane hydration water during depolarization (*Didier et al., 2018*).

Several challenges and considerations in this study warrants discussion. First, maintaining the desired environment (37 °C and 5% $CO_2$) for in vitro cells during MRI scans is challenging. In addition, intracellular accumulation of $Ba^{2+}$ in the $Ba^{2+}$-induced depolarization experiment may affect cellular integrity. In this study, high cell viability (>97.6%) was confirmed using cell viability assays under experimental conditions (*Appendix 4—figure 1*). Second, differences in $T_2$ value among the extracellular media may bias $T_2$ measurements due to the partial volume effect. However, in this study, the differences in $T_2$ among the extracellular media were found to be negligible compared to the observed $T_2$ changes (*Appendix 2—figure 4*). Third, while our findings show that membrane potential-modulating ionic solutions can affect MR parameters, it is important to note that these changes do not measure the membrane potential itself. Fourth, other factors such as pH, energy depletion, or extracellular osmolarity may affect MR parameters by altering cell volume. To minimize the effects of these other contributors, we matched osmolarity across all conditions, provided sufficient glucose to prevent energy depletion, and regulated pH levels by buffering with HEPES. Additionally, to mitigate possible changes in intra/extracellular volume fraction changes caused by cell movements, potentially due to agitation, we centrifuged the cells and imaged the bottom portion of the cell pellet, where cell movement was restricted due to close packing. Fifth, in the in vivo study, we attempted to suppress hemodynamic responses through pharmacological means, using a combination of $N_\omega$-Nitro-L-arginine and nifedipine, both of which are known to inhibit hemodynamic responses in different ways (*Dreier et al., 1995*; *Redmond et al., 2002*), but their effects were not directly confirmed in this study. Future studies that simultaneously evaluate hemodynamic responses would strengthen our conclusions. Finally, our experimental paradigm was based on clamping the membrane potential at a specific level, thus measuring changes in $T_2$ and PSR during static depolarization or hyperpolarization rather than dynamic changes such as those seen during action potentials. Future research could explore temporally varying membrane potential to evaluate the dynamic correlation between membrane potential and MR parameters with high temporal resolution MRI (*Sundaram et al., 2010*; *Toi et al., 2022*).

In summary, our study demonstrates that MR parameters such as $T_2$ relaxation time can detect responses to membrane potential-modulating ionic solutions both in vitro and in vivo. This finding proposes a potential approach for noninvasively detecting changes in membrane potential using MRI.

# Materials and methods

**Key resources table**

| Reagent type (species) or resource | Designation | Source or reference | Identifiers | Additional information |
|---|---|---|---|---|
| Strain, strain background (*Rattus norvegicus*) | Wistar | Orient Bio | Cat #: CrlOri:WI; RRID:RGD_13508588 | |
| Cell line (*Homo sapiens*) | SH-SY5Y | American Type Cell Collection | Cat #: CRL-2266; RRID:CVCL_0019 | |
| Cell line (*Homo sapiens*) | Jurkat E6.1 | American Type Cell Collection | Cat #: TIB-152; RRID:CVCL_0367 | |

## In vitro cell culture

Two human cell lines were utilized for the experiments: SH-SY5Y, an immortalized neuroblastoma line, and Jurkat, a leukemia cell line. Both cell lines were sourced from the American Type Culture

Collection (ATCC), where their identities were authenticated by STR analysis and confirmed to be negative for mycoplasma contamination, as documented in the certificate of analysis provided at the time of purchase. The SH-SY5Y cells were cultured in DMEM/F12 medium supplemented with 10% (v/v) fetal bovine serum (FBS) and 100 U/ml penicillin/streptomycin. The cells were maintained at a constant temperature of 37 °C in a humidified atmosphere containing 5% $CO_2$. Similarly, the Jurkat cells were cultured in RPMI-1640 medium, also supplemented with 10% (v/v) FBS and 100 U/ml penicillin/streptomycin, under the same conditions of temperature and $CO_2$ concentration.

## In vitro manipulation of membrane potential with extracellular media

The baseline extracellular medium was prepared with the following components: KCl = 4.2 mM; NaCl = 145.8 mM; HEPES = 20 mM; glucose = 4.5 g/l; EGTA = 10 μM; pH = 7.2. This baseline medium was considered a control condition for various extracellular media used to adjust membrane potential. Two extracellular media with low $K^+$ concentrations (KCl = 0.2 and 1 mM) were prepared to hyperpolarize the membrane potential. Three extracellular media with high $K^+$ concentrations (KCl = 20, 40, and 80 mM) were prepared to depolarize the membrane potential. A $Ba^{2+}$ medium containing 10 mM $BaCl_2$ was also prepared to depolarize the membrane potential in a different way, i.e., as a $K^+$ channel blocker. These seven extracellular media were used for both MR imaging and patch clamp recording in vitro. Sodium ion concentrations ($[Na^+]$) in all media were controlled to match the osmolarity with the baseline medium. The composition of all extracellular media is detailed in *Appendix 1—table 3*.

## Preparation of cells for in vitro MR measurement

SH-SY5Y cells were dissociated from their culture plates using 0.5 mM EDTA solution, then concentrated into a pellet (~70 μl) by centrifugation at 250×g for two minutes. The pellet was resuspended in the culture medium and divided evenly into seven aliquots. Each cell suspension was centrifuged and resuspended in each of the seven different media specified in *Appendix 1—table 3*. Centrifugation and resuspension were repeated three more times to completely clear out the culture medium. Each cell suspension with a different extracellular medium was then loaded into two wells on the same column of an acrylic container with 14 wells (matrix = 2 × 7) and centrifuged again to concentrate into pellets (*Figure 1*). The acrylic container was purposely formed into a spherical segment to enhance the homogeneity of the static magnetic field (*Lee et al., 2020*). The preparation of Jurkat cell samples was the same as for the SH-SY5Y cell samples. The preparation required 40–60 min, followed by an incubation period of 20–30 min.

## In vitro MRI experiment

In vitro MRI experiments were performed on a 9.4T MRI system (BioSpec 94/30 USR, Bruker BioSpin) at room temperature. A volume coil with an inner diameter of 86 mm was utilized for both radiofrequency (RF) pulse transmission and signal reception. Within the acrylic container, two wells on the same column (matrix = 2 × 7) contained identical cell pellets with the same extracellular media. MRI signals from seven different cell samples in the horizontal direction were separated by one-dimensional frequency encoding along that direction (*Figure 1*). The MRI pulse sequence employed for mapping the $T_2$ value was a single-echo spin-echo (SESE) sequence with 50 variable echo times (TE) spaced between 9.5 and 290.5ms on a logarithmic scale. For mapping the MT parameters, an inversion recovery multi-echo spin-echo (IR-MESE) sequence was used. The inversion times (TI) for the IR-MESE sequence were optimized using the theory of Cramér-Rao lower bounds (*Li et al., 2010*). The optimized TIs ranged from 4 to 10,079.4ms. After each TI, 16 spin-echo trains were acquired with an echo spacing of 9.5ms. Total scan time for both sequences was 23 min. Experiments were repeated 15 times for SH-SY5Y cells and 7 times for Jurkat cells, replacing cells in each repetition. Other scan parameters are detailed in *Appendix 1—table 1*.

## Animals

Male Wistar rats aged 8 weeks (250–300 g, Orient Bio) were used for MRI experiments after undergoing a craniotomy. All animal experiments were approved by the Institutional Animal Care and Use Committee at the National Cancer Center Korea (NCC-22–740). The rats were housed in ventilated cages under a 12 hr/12 hr light/dark cycle and provided with ad libitum access to food and water.

## In vivo manipulation of membrane potential with artificial cerebrospinal fluid (aCSF)

The membrane potential of the exposed cortex was manipulated by directly perfusing the region of interest of the cerebral cortex with aCSF after a craniotomy. The baseline aCSF was prepared with the following components: KCl = 3 mM; NaCl = 135 mM; $MgCl_2$ = 3 mM; HEPES = 20 mM; glucose = 4.5 g/l; EGTA = 2 mM; $N_\omega$-Nitro-L-arginine=1 mM; Nifedipine = 0.1 mM; pH = 7.4. Hemodynamic effects were pharmacologically suppressed using $N_\omega$-Nitro-L-arginine, Nifedipine, and EGTA. $N_\omega$-Nitro-L-arginine suppresses depolarization-induced hemodynamic response by blocking the synthesis of nitric oxide, which acts as a vasodilator (*Dreier et al., 1995*). Nifedipine blocks voltage-sensitive $Ca^{2+}$ channels, and EGTA chelates free $Ca^{2+}$ to inhibit the hemodynamic response (*Redmond et al., 2002*). To induce depolarization, aCSF with high [$K^+$] (KCl = 40 and 80 mM) was prepared. The osmolarity of the aCSF was matched with the concentration of NaCl.

## Rat surgery

A craniotomy was performed on a Wistar rat. The experimental setup after a surgical procedure is illustrated in *Figure 5*. The surgery was performed following an established protocol (*Mostany and Portera-Cailliau, 2008*) alike to that used in other MRI studies (*Stanisz et al., 2002*; *Autio et al., 2014*). Anesthesia was induced with 3% isoflurane in $O_2$ and maintained with 2–3% isoflurane during the surgical procedure. Body temperature was maintained at 36.5–37.5°C with an infrared lamp. The head was fixed with a small animal stereotaxic frame. The hair on the scalp was shaved with a veterinary clipper. The skin and periosteum over the skull were removed using a scalpel and surgical scissors. A 3.0-mm-diameter burr hole was opened using a dental drill with its center at the coordinates of 2.5 mm anterior and 2.0 mm lateral to the lambda. Then, a cylindrical chamber was implanted upon the burr hole with cyanoacrylate glue and dental composite resin. The chamber was filled with the baseline aCSF and connected to inlet and outlet perfusion tubes. The exposed cerebral cortex inside the chamber was perfused with the baseline aCSF at a flow rate of 0.6 ml/min using peristaltic pumps.

## In vivo MRI experiment

The rat with a cranial chamber installed on the cortical surface was placed on a 7T MRI system (BioSpec 70/20 USR, Bruker BioSpin), fixed in a customized cradle with two ear-bars and a bite-bar. A customized surface coil (rectangular, 35 mm × 20 mm) was used for RF pulse transmission and signal reception. Body temperature was maintained at 36.5–37.5°C using a warm air blower. Anesthesia was maintained with 2% isoflurane in $O_2$ (0.6 l/min). Respiration rate and body temperature were monitored throughout the MRI experiment. MR images were acquired in a 2 mm coronal slice through the center of the burr hole (*Figure 5*), using a multi-echo spin-echo (MESE) sequence with 20 TEs (7.5–150ms). Other scan parameters are detailed in *Appendix 1—table 2*.

A total of seven rats were subjected to four sequential experimental conditions, as depicted in *Figure 5*. First, the exposed cerebral cortex was perfused with baseline aCSF ([$K^+$]=3 mM). Second, as a depolarizing condition, the perfusion media was switched to depolarizing aCSF of [$K^+$]=40 mM. Third, membrane potential was further depolarized by perfusion with aCSF of [$K^+$]=80 mM. Finally, as a recovery condition, the perfused aCSF was changed back to baseline aCSF. During each condition, MR images were acquired with MESE sequences for 12 min. Two rats underwent the whole four conditions, while five other rats did not undergo the recovery condition. As a control experiment, another set of rats (n=5) underwent perfusion with the baseline aCSF for the same duration (48 min) as the previous experiment, and MR images were acquired with four MESE sequences, each for 12 min. Throughout the experiment, the perfusion rate was maintained at 0.6 ml/min.

## Quantification of MR parameters

For the in vivo MR images acquired with a MESE sequence, echo trains were matched with a simulated dictionary of decay curves of multi-echo spin-echo signals created with the stimulated echo and slice profile correction (*McPhee and Wilman, 2017*) to estimate $T_2$ values. For the in vitro one-dimensional MR images acquired with the SESE sequence, signals were fitted to a mono-exponential function to estimate $T_2$ values. For the in vitro one-dimensional MR images acquired with the IR-MESE sequence, signals were fitted to a bi-exponential function (*Edzes and Samulski, 1977*; *Gochberg and Gore,*

*2003*) to estimate MT parameters. 16 spin echoes acquired after each TI were averaged to improve SNR. The MT parameters such as PSR and $k_{mf}$ were derived from this fitting process.

A 1ms hard inversion pulse selectively inverted the magnetization of free water protons, leading to cross-relaxation between the longitudinal magnetization of free water protons ($M_{z,f}$) and macromolecular protons ($M_{z,m}$). This interaction resulted in a bi-exponential magnetization recovery characterized by a fast longitudinal relaxation rate $R_1^+$ and a slow relaxation rate $R_1^-$ (=1 /$_{T1}$), with the latter corresponding to the conventional spin-lattice relaxation rate (*Gochberg and Gore, 2003*; *Xu et al., 2014*):

$$\frac{M_{z,f}(t)}{M_{\infty,f}} = b_f^{\ +} \exp(-R_1^+ t) + b_f^{\ -} \exp(-R_1^- t) + 1 \tag{4}$$

where $M_{\infty,f}$ denotes the equilibrium magnetization of free water protons, and $b_f^+$ and $b_f^-$ denote the amplitudes for the exponential terms associated with $R_1^+$ and $R_1^-$, respectively. By fitting this bi-exponential model to the inversion recovery signals using least squares, estimates of $R_1^+$, $R_1^-$, $b_f^+$, and $b_f^-$ were obtained. These parameters are related to the MT parameters, PSR and $k_{mf}$, by the following equations:

$$2R_1^\pm = R_{1,f} + R_{1,m} + k_{fm} + k_{mf} \pm \sqrt{\left(R_{1,f} - R_{1,m} + k_{fm} - k_{mf}\right)^2 + 4k_{fm}k_{mf}} \tag{5}$$

$$b_f^\pm = \pm \frac{\left(R_{1,f} - R_1^\mp\right)\left(\frac{M_{z,f}(0)}{M_{\infty,f}} - 1\right) + k_{fm}\left(\frac{M_{z,f}(0)}{M_{\infty,f}} - \frac{M_{z,m}(0)}{M_{\infty,m}}\right)}{R_1^+ - R_1^-} \tag{6}$$

$$PSR = k_{fm}/k_{mf} \tag{7}$$

where $R_{1,f}$ and $R_{1,m}$ denote the longitudinal relaxation rates of free water and macromolecular protons, respectively, in the absence of cross-relaxation. $k_{fm}$ and $k_{mf}$ denote the magnetization transfer rates from free water to macromolecules and vice versa, respectively, $M_{z,m}(0)$ denotes the longitudinal magnetization of macromolecular protons immediately after the inversion pulse. $M_{\infty,m}$ denotes the equilibrium magnetization of macromolecular protons. According to previous studies (*Gochberg and Gore, 2003*; *Gochberg et al., 1999*), $M_{z,m}(0)/M_{\infty,m}$ can be determined numerically by the Bloch equations. Assuming $R_{1,f} = R_{1,m}$ (*Li et al., 2010*; *Cabana et al., 2015*), the equations [*Equations 5–7*] can be simplified to explicitly calculate PSR and $k_{mf}$.

$$PSR = \frac{b_f^{\ +}}{b_f^{\ -} - \frac{M_{z,m}(0)}{M_{\infty,m}} + 1} \tag{8}$$

$$k_{mf} = \frac{R_1^+ - R_1^-}{1 + PSR} \tag{9}$$

## Patch clamp recordings

The membrane potential of SH-SY5Y cells was recorded at room temperature using the whole-cell mode of the patch clamp technique (*Hamill et al., 1981*). The bath solution was the same as the extracellular medium used in the MRI experiment and was constantly perfused at a flow rate of 2 ml/min. The composition of the pipette solution was as follows: KCl = 140 mM; NaCl = 5 mM; MgCl2=3 mM; HEPES = 10 mM; Mg-ATP=1 mM; Na-GTP=0.5 mM. The pH of the pipette solution was adjusted to 7.4 using KOH. Calcium ions ($Ca^{2+}$) were not included in the pipette solution to minimize $Ca^{2+}$-dependent currents. The resistance of the electrode was 3–5 MΩ with the internal solution filled. Recordings were performed with a patch amplifier (Axopatch-1D; Axon Instruments) and a current clamp was also used to monitor the membrane potential. The experiment was repeated three times, with cells replaced at each repetition.

## Acknowledgements

We thank Dr. Silvia Mangia for insightful discussions on the role of hydration water in functional MRI. Animal molecular imaging facility at the National Cancer Center Korea contributed to supportive animal magnetic resonance imaging. We thank Minsun Kim and Soyeon Jeon (National Cancer Center) for helping with rat craniotomies and MRI experiments. This work was supported by the National Research Foundation of Korea grant funded by the Ministry of Science and ICT (NRF-2019M3C7A1031993, NRF-2019M3C7A1031994, and NRF-2023R1A2C3007075). This work was supported by the Commercialization Promotion Agency for R&D Outcomes (COMPA) funded by the Ministry of Science and ICT (NTIS1711198890).

## Additional information

### Funding

| Funder | Grant reference number | Author |
|---|---|---|
| National Research Foundation of Korea | 2019M3C7A1031993 | Phan Tan Toi Jang-Yeon Park |
| National Research Foundation of Korea | 2019M3C7A1031994 | Kyeongseon Min Jongho Lee |
| National Research Foundation of Korea | 2023R1A2C3007075 | Jang-Yeon Park |
| Commercializations Promotion Agency for R and D Outcomes | NTIS1711198890 | Daehong Kim |

The funders had no role in study design, data collection and interpretation, or the decision to submit the work for publication.

### Author contributions

Kyeongseon Min, Conceptualization, Investigation, Visualization, Methodology, Writing – original draft, Writing – review and editing; Sungkwon Chung, Resources, Investigation, Visualization, Methodology, Writing – review and editing; Seung-Kyun Lee, Conceptualization, Methodology, Writing – review and editing; Jongho Lee, Resources, Supervision, Methodology, Writing – review and editing; Phan Tan Toi, Investigation, Methodology, Writing – review and editing; Daehong Kim, Resources, Investigation, Methodology, Writing – review and editing; Jung Seung Lee, Resources, Methodology, Writing – review and editing; Jang-Yeon Park, Conceptualization, Resources, Supervision, Methodology, Writing – original draft, Writing – review and editing

### Author ORCIDs

Kyeongseon Min http://orcid.org/0000-0001-7614-2089
Sungkwon Chung https://orcid.org/0000-0003-0505-8773
Jang-Yeon Park https://orcid.org/0000-0002-0586-1606

### Ethics

All animal experiments were approved by the Institutional Animal Care and Use Committee at the National Cancer Center Korea (NCC-22-740).

Reviewer #1 (Public review): https://doi.org/10.7554/eLife.101642.3.sa1
Reviewer #2 (Public review): https://doi.org/10.7554/eLife.101642.3.sa2
Author response https://doi.org/10.7554/eLife.101642.3.sa3

## Additional files

### Supplementary files

MDAR checklist

Source data 1. This source data was used for generating the figures in this article.

### Data availability

The source data used to generate the figures are included in Source data 1.

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

# Appendix 1

This appendix provides detailed information on the experimental parameters and conditions used in both in vitro and in vivo MRI studies, as well as the composition of extracellular media applied to modulate cellular membrane potential.

**Appendix 1—table 1.** The scan parameters of the sequences used in the in vitro MR experiments. The cell samples were scanned with two types of sequences, single-echo spin-echo (SESE) and inversion recovery multi-echo spin-echo (IR-MESE) sequences, on a 9.4 T MRI. $T_2$ was estimated from the SESE sequence. $T_1$, PSR, and $k_{mf}$ were estimated from the IR-MESE sequence, and its inversion times (TIs) were optimized using the theory of Cramér-Rao low bounds: TIs = 4, 4, 4, 4, 4, 4, 4, 4, 17.91, 18.18, 18.18, 18.2, 18.2, 18.21, 18.21, 18.24, 18.24, 18.24, 18.24, 18.24, 18.31, 18.31, 18.32, 18.46, 55.18, 55.26, 55.53, 55.98, 163.5, 164.83, 164.92, 164.93, 164.96, 165.25, 165.25, 165.62, 197.68, 1976.77, 2280.95, and 10076.4 ms. The recovery time or repetition time was set long enough to ensure full relaxation of nuclear magnetization.

| Sequence type | SESE | IR-MESE |
|---|---|---|
| Recovery time (ms) | N/A | 15000 |
| Repetition time (ms) | 15000 | N/A |
| Inversion time (ms) | N/A | 4–10,079.4 (40 steps) |
| Echo time (ms) | 9.5–290.5 (linear, 50 steps) | 9.5–152 (linear, 16 steps) |
| Resolution (mm) | 0.5 | |
| Slice thickness (mm) | 1 | |
| Estimated parameters | $T_2$ | $T_1$, PSR, and $k_{mf}$ |

**Appendix 1—table 2.** The scan parameters of the sequence used in the in vivo rat MR experiments. The rats were scanned with a multi-echo spin-echo (MESE) sequence on a 7 T MRI.

| Sequence type | MESE |
|---|---|
| Repetition time (ms) | 1000 |
| Inversion time (ms) | N/A |
| Echo time (ms) | 7.5–150 (linear, 20 steps) |
| Resolution (mm$^2$) | 0.3×0.3 |
| FOV (mm$^2$) | 28.8×28.8 |
| Slice thickness (mm) | 2 |
| Estimated parameters | $T_2$ |

**Appendix 1—table 3.** The composition of the extracellular media used to modulate membrane potential in vitro.
The sodium chloride concentrations were adjusted to maintain the same osmolarity across all media. Besides the inorganic salts listed in this table, all media commonly contained HEPES = 20 mM; glucose = 4.5 g/l; EGTA = 10 µM; pH = 7.2.

| Medium type | KCl (mM) | BaCl$_2$ (mM) | NaCl (mM) |
|---|---|---|---|
| Baseline | 4.2 | | 145.8 |
| | 0.2 | | 149.8 |
| Low [K$^+$] | 1 | | 149 |
| | 20 | | 130 |
| | 40 | | 110 |
| High [K$^+$] | 80 | | 70 |
| [Ba$^{2+}$] | 4.2 | 10 | 130.8 |

## Appendix 2

### Membrane potential-induced changes in $T_1$: SH-SY5Y cells

The changes of $T_1$ in SH-SY5Y cells when the membrane potential ($V_m$) was modulated by varying $[K^+]$ = 0.2–80 mM or adding $[Ba^{2+}]$ = 10 mM are shown in *Appendix 2—figures 1 and 2*. A linear mixed-effect model was employed to analyze the effect of changes in $V_m$ ($\Delta V_m$) on $T_1$. This model included $T_1$ as a dependent variable, $\Delta V_m$ and its interaction with a group variable (indicating whether $\Delta V_m$ is induced by $[Ba^{2+}]$ or $[K^+]$) as fixed effects, and cell batch as a random effect. The analysis revealed no significant interaction term ($p$ = 0.188), showing that changes in $T_1$ do not depend on the specific method of membrane potential modulation. The model yielded the following relationship:

$$T_1 \, (\text{ms}) = 1.94 \times 10^3 \left( 1 + 0.00197 \, \Delta V_m \right) \tag{A2.1}$$

The effect $\Delta V_m$ on $T_1$ was significant (p < 0.0001), indicating an increase in $T_1$ during depolarization and a decrease during hyperpolarization. This trend is the same with $T_2$ regarding membrane potential, while the sensitivity is 3.5 times smaller compared to $T_2$. Subsequent post-hoc analysis compared each experimental condition to the control using Dunnett's test to account for multiple comparisons, showing significant changes in $T_1$ across all conditions tested (p < 0.0001).

### Membrane potential-induced changes in $T_1$: Jurkat cells

The changes in $T_1$ were also evaluated using a different cell line, the Jurkat, under the same experimental conditions applied to SH-SY5Y cells. Each experimental condition was compared to the control using Dunnett's test to account for multiple comparisons (*Appendix 2—figure 3*). As the same with observation in SH-SY5Y cells, Jurkat cells showed positive changes in $T_1$ during hyperpolarization and negative changes during depolarization, indicating that the detectability of membrane potential changes by $T_1$ does not appear specific to cell type.

### Quantification of $T_1$ and $T_2$ of extracellular media

To estimate the $T_1$ and $T_2$ of extracellular media specified in *Appendix 1—table 3*, 200 μl of each media sample was loaded into two wells on the same column of the acrylic container with 14 wells (*Figure 1A*) and placed in a 9.4 T preclinical MRI system (BioSpec 94/30 USR, Bruker BioSpin). $T_1$ and $T_2$ were measured using an adiabatic inversion recovery multi-echo spin-echo sequence. 20 variable inversion times (TI) were spaced between 600 and 15,000 ms on a logarithmic scale. After each TI, 50 multi-echo spin-echo trains were acquired with an echo spacing of 9.5 ms (echo time (TE) = 9.5 to 475 ms). For the $T_1$ estimation, the 50 multi-echo spin-echo signals acquired after each TI were averaged and fitted to a mono-exponential function:

$$S(t) = A + B \exp \left( -\frac{t}{T_1} \right) \tag{A2.2}$$

where $S(t)$ denotes the averaged signal after an inversion time of $t$, and A and B denote the coefficients of the fitting. For $T_2$ estimation, the 20 spin-echo signals at the same TE but different 20 TIs were averaged. These averaged 50 multi-echo spin-echo trains were then matched with a simulated dictionary of decay curves. This dictionary was constructed from simulation of multi-echo spin-echo signals, incorporating corrections for stimulated echoes and slice profile effects (*McPhee and Wilman, 2017*).

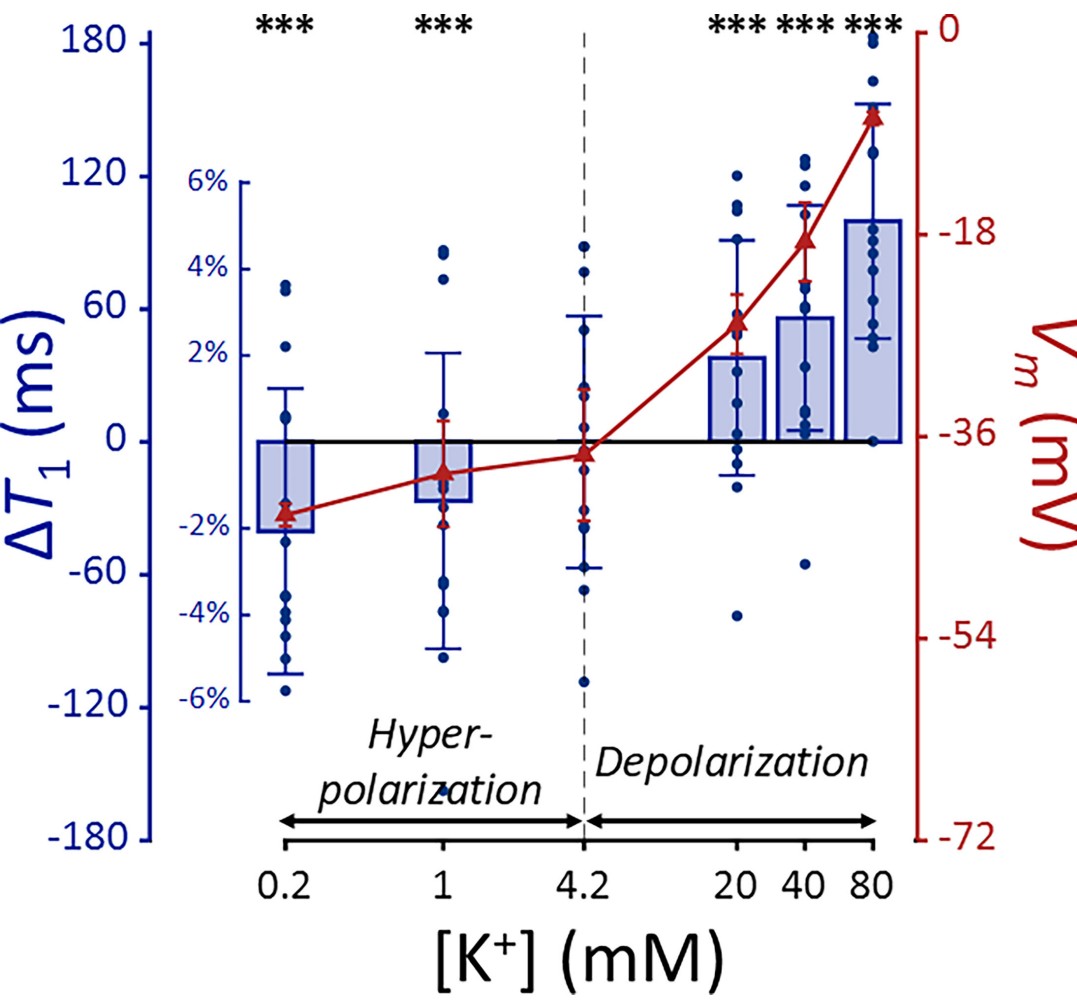

**Appendix 2—figure 1.** $T_1$ and membrane potential ($V_m$) of SH-SY5Y cells versus extracellular $K^+$ concentrations ($[K^+]$). Changes in $T_1$ are displayed with blue bars (n = 15). Membrane potentials are plotted with red triangles (n = 3). The abscissa is in logarithmic scale. Error bars denote standard deviation. Statistical significance of changes in T1 is marked with asterisks (***: p < 0.001).

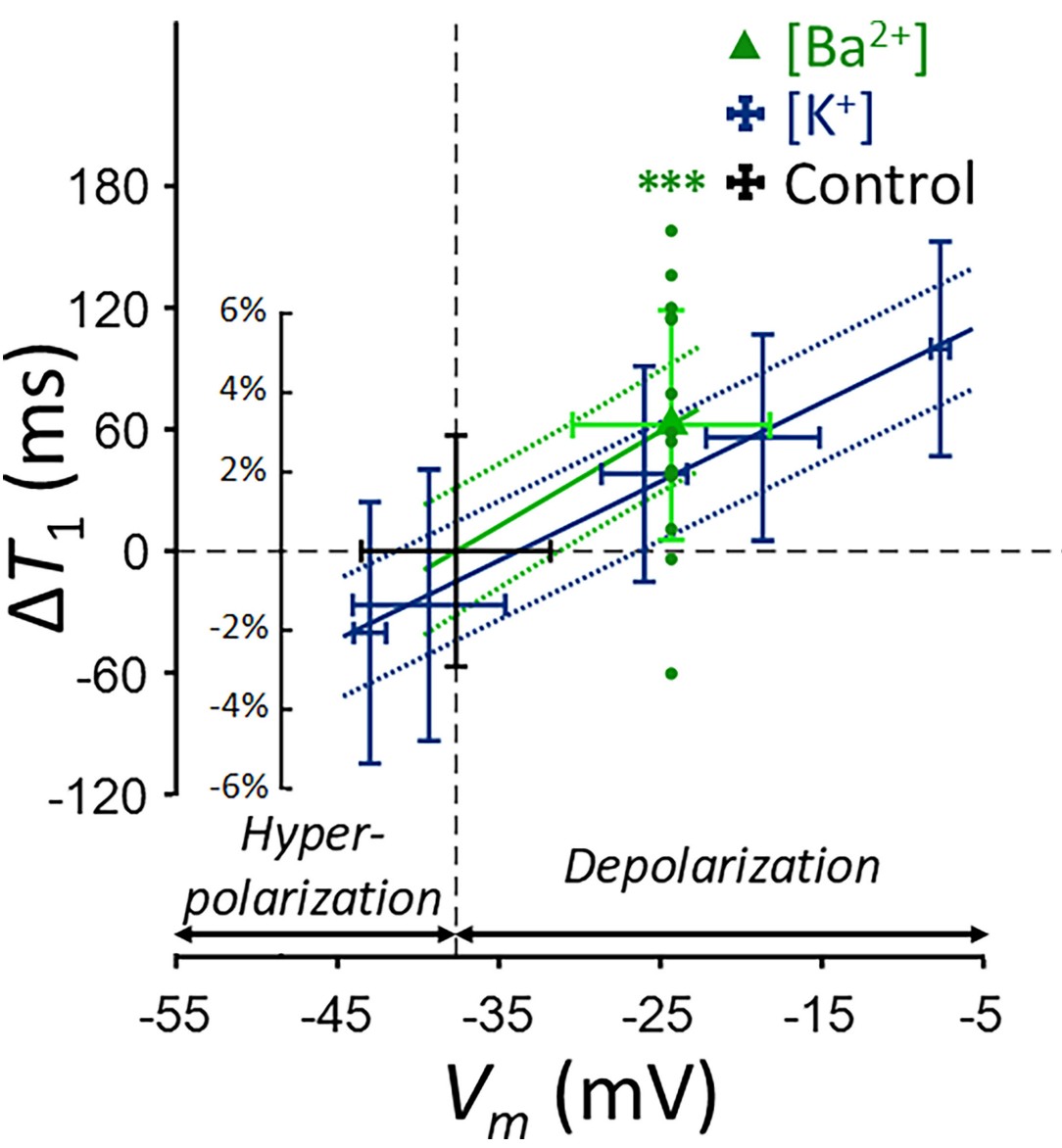

**Appendix 2—figure 2.** Changes in $T_1$ of SH-SY5Y cells across experimental conditions: [K$^+$] = 0.2–80 mM (blue cross) and [Ba$^{2+}$] = 10 mM (green triangle), compared to the control condition (black cross). Data from fifteen experiments (n = 15) are displayed. Linear regression lines for [K$^+$] data (blue solid line) and [Ba$^{2+}$] data (green solid line) are drawn along with 95% confidence intervals. Error bars denote standard deviation. Statistical significance of changes in $T_1$ with [Ba$^{2+}$] = 10 mM is marked with asterisks (***: $p < 0.001$).

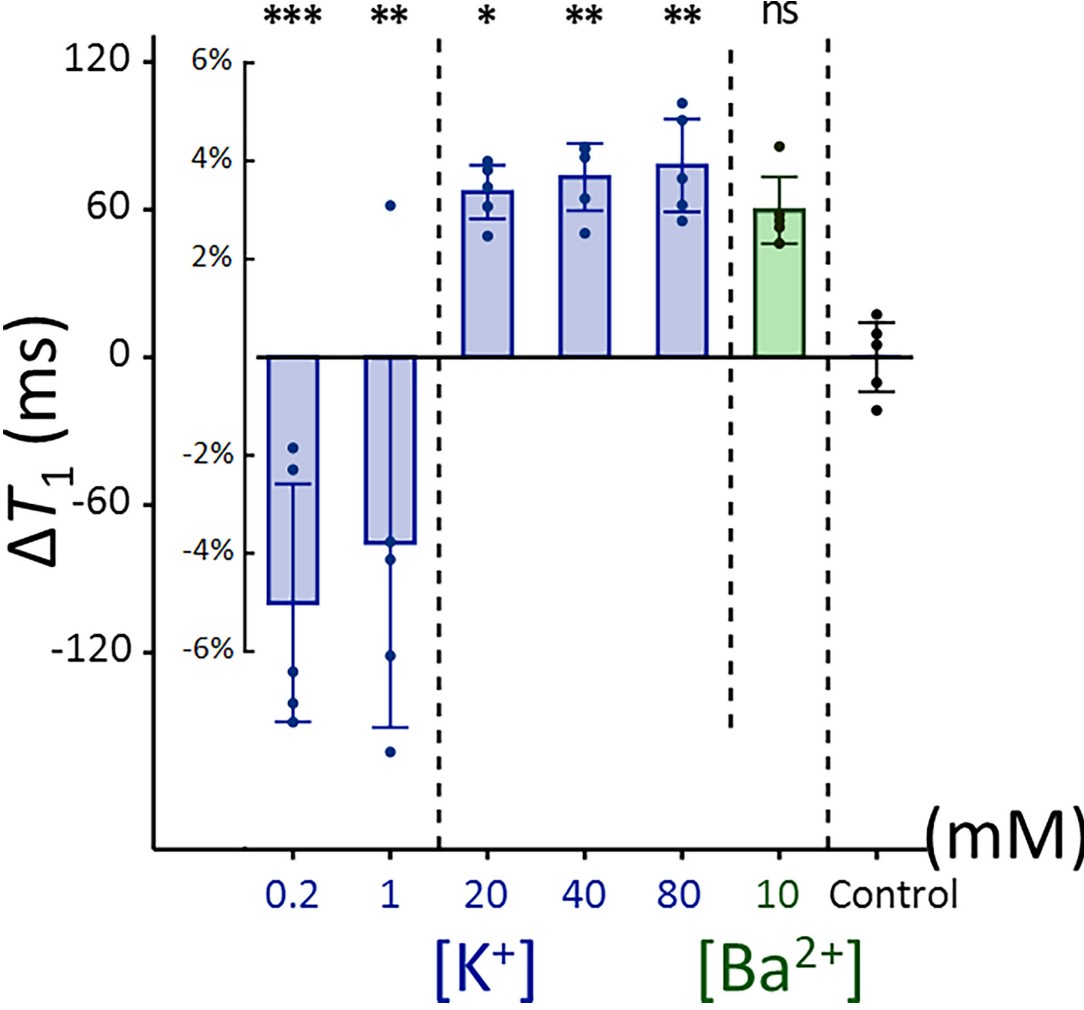

**Appendix 2—figure 3.** Changes in $T_1$ of Jurkat cells across experimental conditions: $[K^+]$ = 0.2–80 mM (blue bar) and $[Ba^{2+}]$ = 10 mM (green bar), compared to the control condition of $[K^+]$ = 4.2 mM (n = 5). Error bars denote standard deviation. Statistical significance of changes in $T_1$ is marked with asterisks (ns: $p > 0.05$, *: $p < 0.05$, **: $p < 0.01$, ***: $p < 0.001$).

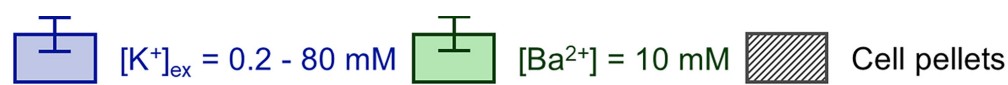

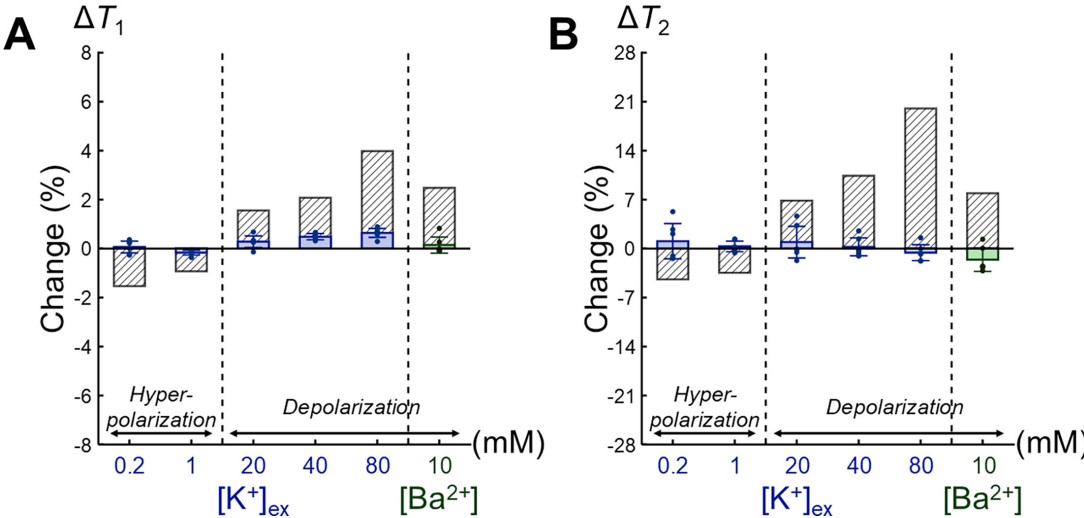

**Appendix 2—figure 4.** Comparison of changes in relaxation times between extracellular media and SH-SY5Y cell pellets. Changes in (A) $T_1$ and (B) $T_2$ are expressed as percentage changes relative to the control condition of $[K^+] = 4.2$ mM. The results for extracellular media are displayed as blue bars for the extracellular $K^+$ concentrations ($[K^+]$) and green bars for the $Ba^{2+}$ concentration ($[Ba^{2+}]$) (n = 6). For comparison, the results for SH-SY5Y cell pellets corresponding to each extracellular medium are plotted. Error bars denote standard deviation.

## Appendix 3

### Quantitative assessment of in vivo T$_2$ mapping

In this study, in vivo $T_2$ mapping was conducted using a multi-echo spin-echo (MESE) sequence. MESE signals exhibit nonexponential decay patterns due to stimulated echoes and imperfect slice profiles (**Hennig, 1991**). To correct for these effects, the MESE signals were fitted to a simulated dictionary of MESE signals, which was created using the extended phase graph (EPG) method (**McPhee and Wilman, 2017**). This dictionary included 360,000 MESE signal curves, spanning $T_2$ ranges from 20 to 200 ms and $B_1$ ranges from 0.5 to 1.5.

**Appendix 3—figure 1** illustrates the procedure to assess the SNR and fitting quality for $T_2$ mapping using an example MESE image from the in vivo rat experiment. First, the background noise of the MESE image was calculated. When a complex MR image contains Gaussian noise with a standard deviation of $\sigma$, the resulting magnitude image will exhibit Rician noise. This can be approximated as Gaussian noise if the SNR (= $A/\sigma$) is greater than 3, where $A$ is the magnitude of signal in the absence of noise (**Gudbjartsson and Patz, 1995**). Rician noise was sampled from 64 voxels in the background of the magnitude image and the noise level $\sigma$ was calculated from the average of the sampled Rician noise $M$ using the formulaxels in the background of the magnitude (**Gudbjartsson and Patz, 1995**):

$$M = \sigma \sqrt{\frac{\pi}{2}} \tag{A3.1}$$

Subsequently, a mask was created by thresholding the magnitude image at SNR > 3. $T_2$ mapping was performed by fitting the MESE signal to the simulated dictionary across the masked region. The fit quality was assessed using the normalized root mean squared error (NRMSE) and adjusted $R^2$, calculated as follows:

$$\text{NRMSE} = \frac{\sum (y_i - f_i)^2}{\sum y_i^2} \tag{A3.2}$$

$$\text{Adjusted } R^2 = 1 - \frac{n-1}{n-p-1} \frac{\sum (y_i - f_i)^2}{\sum (y_i - \bar{y})^2} \tag{A3.3}$$

where is the mean of the signals, $y_i$ is the $i$th signal, $f_i$ is the $i$th fitted value, $n$ is the sample size (number of echoes), and $p$ is the number of variables.

Finally, the magnitude signal in the ROI beneath the perfusion chamber (width = 1.8 mm, depth = 0.6 mm) was analyzed. After averaging the magnitude signals over the ROI, it is confirmed that the signal is well above the noise floor ($3\sigma$). After performing $T_2$ fitting, the fit quality was examined with NRMSE and adjusted $R^2$.

This analysis was conducted across all seven in vivo rat models. **Source data 1** contains the fitted $T_2$ and $B_1$ value, NRMSE and adjusted $R^2$ value, the signal intensity of the last echo in the ROI, and the noise level $\sigma$. Notably, the maximum NRMSE was 0.043 and the minimum adjusted $R^2$ was 0.995, indicating a high degree of alignment between the observed magnitude signals and the fitting curves.

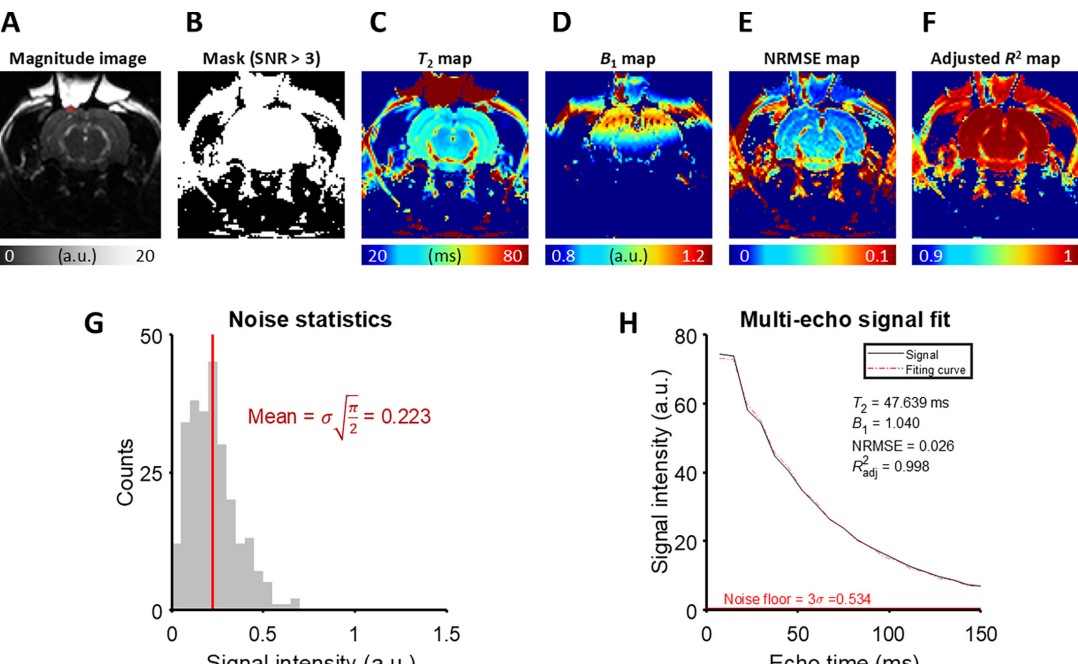

**Appendix 3—figure 1.** The detailed analysis of $T_2$ fitting on an example MESE image from the in vivo rat experiment. (**A**) A magnitude image at the last echo time. The ROI is marked with a red rectangle. (**B**) A mask generated by thresholding the magnitude image with SNR > 3. (**C**) $T_2$ map, (**D**) $B_1$ map, (**E**) NRMSE map, and (**F**) Adjusted $R^2$ map produced by the fitting procedure. (**G**) The noise statistics of the background of the magnitude image. (**H**) The magnitude signal averaged over the ROI and its fitting curve.

## Appendix 4

### Viability assay of SH-SY5Y cells with live/dead staining

SH-SY5Y cells were cultured, harvested, and centrifuged to obtain a cell pellet. This cell pellet was resuspended in the culture medium and divided into seven equal aliquots. Each aliquot was centrifuged and resuspended three times using one of the seven different media specified in *Appendix 1—table 3*. For the viability assay, staining media were prepared by adding 2 µM calcein-AM and 4 µM Ethidium homodimer-1 to each of the seven media. The cell suspensions were then centrifuged and resuspended again using the staining media. Following this, each cell suspension was transferred to separate wells in a chambered coverglass and incubated at room temperature for 20 minutes. After another round of centrifugation, cell pellets were obtained and analyzed using a confocal microscope (Leica TCS SP8 STED, Leica Microsystems). Cell viability was assessed based on green fluorescence indicating live cells and red fluorescence indicating dead cells, under excitation wavelengths of 488 and 543 nm, respectively. Live and dead cells were manually counted with QuPath (*Bankhead et al., 2017*), and the cell viability was calculated as the ratio of live cells to total cells.

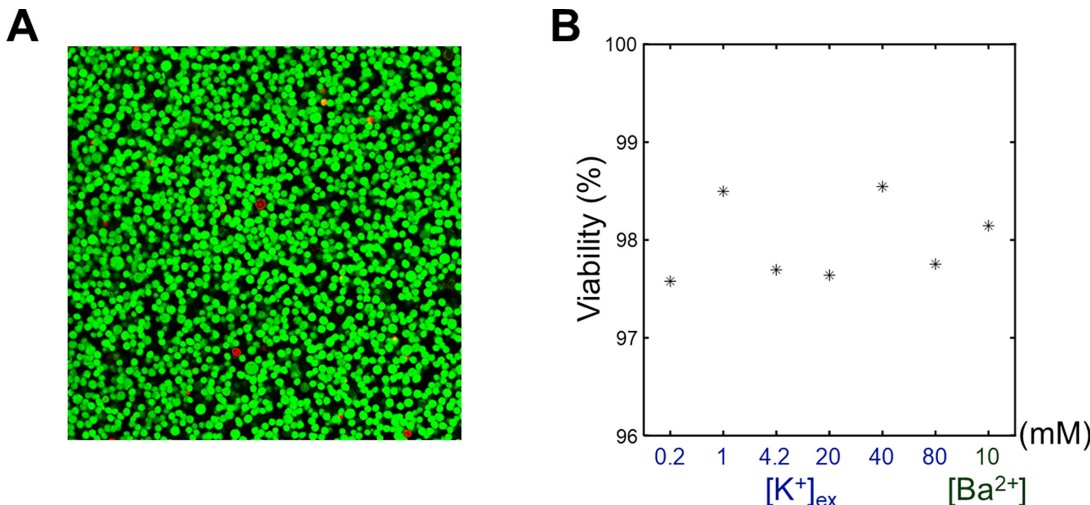

**Appendix 4—figure 1.** The viability assay of SH-SY5Y cells. (**A**) A representative confocal microscopy image of an SH-SY5Y pellet. [K⁺] of extracellular medium was 4.2 mM. Live cells (green) were stained with calcein-AM, and dead cells (red) were stained with EthD-1. (**B**) The viabilities of SH-SY5Y cells versus the extracellular $K^+$ concentrations ($[K^+]$) and $Ba^{2+}$ concentration ($[Ba^{2+}]$).

