## [Editor Report · eLife Assessment]

The authors show MRI relaxation time changes that are claimed to originate from cell membrane potential changes. This would be a substantial contribution if true because it may provide a mechanism whereby membrane potential changes could be inferred noninvasively. However, the membrane potential manipulations applied here are performed on a slow time scale and are known to induce cell swelling. Cell swelling has been previously shown to affect relaxation time. Experiments could be performed to rule out this hypothesis, but the authors have chosen not to perform these experiments. The study is therefore **useful**, but the evidence is **incomplete**.

---

## [Referee Report · Reviewer #1 (Public review)]

Summary:

This paper examines changes in relaxation time (T1 and T2) and magnetization transfer parameters that occur in a model system and in vivo when cells or tissue are depolarized using an equimolar extracellular solution with different concentrations of the depolarizing ion K+. The motivation has been revised to state that the results suggest a potential approach to non-invasively detect changes in membrane potential using MRI.

Strengths:

The authors argue that the use of various concentrations of KCL in the extracellular fluid depolarize or hyperpolarize the cell pellets used, and that this change in membrane potential is the driving force for the T2 (and T1-supplementary material) changes observed. In particular, they report an increase in T2 with increasing KCL concentration in the extracellular fluid (ECF) of pellets of SH-SY5Y cells. To offset the increasing osmolarity of the ECF due to the increase in KCL, the NaCL molarity of the ECF is proportionally reduced. The authors measure the intracellular voltage using patch clamp recordings, which is a gold standard. With 80 mM of KCL in the ECF, a change in T2 of the cell pellets of ~10 ms is observed with the intracellular potential recorded as about -6 mv. A very large T1 increase of ~90 ms is reported under the same conditions. The PSR (ratio of hydrogen protons on macromolecules to free water) decreases by about 10% at this 80 mM KCL concentration. Similar results are seen in a Jurkat cell line and similar, but far smaller changes are observed in vivo, for a variety of reasons discussed. As a final control, T1 and T2 values are measured in the various equimolar KCL solutions. As expected, no significant changes in T1 and T2 of the ECF were observed for these concentrations.

Weaknesses:

While the concepts presented are interesting, and the actual experimental methods seem to be nicely executed, the conclusions are not supported by the data for a number of reasons. This is not to say that the data isn't consistent with the conclusions, but there are other controls not included that would be necessary to draw the conclusion that it is membrane potential that is driving these T1 and T2 changes. The results are consistent with Stroman et al. Magn. Reson. in Med. 59:700-706 (increased T2 with KCL) as well as some other cited work. However all those authors emphasize that cell swelling is the mechanism, not cell membrane potentials.

It is well established that cells swell/shrink upon depolarization/hyperpolarization. Cell swelling is accompanied by increased light transmittance in vivo, and this should be true in the pellet system as well. In a beautiful series of experiments, Stroman et al. (2008) showed in perfused brain slices that the cells swell upon equimolar KCL depolarization and the light transmittance increases. The time course of these changes is quite slow, of the order of many minutes, both for the T2-weighted MRI signal and for the light transmittance. Stroman et al. also show that hypoosmotic changes produce the exact same timecourse as the KCL depolarization changes (and vice versa for the hyperosmotic changes - which cause cell shrinkage). Their conclusion therefore, was that cell swelling (not membrane potential) was the cause of the T2-weighted changes observed, and that these were relatively slow (on the scale of many minutes).

What are the implications for the current study? Well, for one, the authors cannot exclude cell swelling as the mechanism for T2 changes, as they have not measured that. It is however well established that cell swelling occurs during depolarization, so this is not in question. Water in the pelletized cells is in slow/intermediate exchange with the ECF, and the solutions for the two compartment relaxation model for this are well established see Menon and Allen, Magn. Reson. in Med. 20:214-227, 1991. The T2 relaxation times should be multiexponential (see point (3) further below). The current work cannot exclude cell swelling as the mechanism for T2 changes (it is mentioned in the paper, but not dealt with). Water entering cells dilutes the protein structures, changes rotational correlation times of the proteins in the cell and is known to increase T2. The PSR confirms that this is indeed happening, so the data in this work is completely consistent with the Stroman work and completely consistent with cell swelling associated with depolarization. The authors should have performed light scattering studies to demonstrate the degree cell swelling or shrinkage. Measuring intracellular potential is not enough to clarify the mechanism.

So why does it matter whether the mechanism is cell swelling or membrane potential? The reason is response time. Cell swelling due to depolarization is a slow process, slower than hemodynamic responses that characterize BOLD. And in fact, cell swelling under normal homeostatic conditions in vivo is virtually non-existent. Only sustained depolarization events typically associated with non-naturalistic stimuli or brain dysfunction produce cell swelling. Membrane potential changes associated with neural activity, on the other hand, are very fast. In this manuscript, the authors have convincingly shown a signal change that is virtually the same as what was seem in the Stroman publication, but they have not shown that there is a response that can be detected with anything approaching the timescale of an action potential. So one cannot definitely say that the changes observed are due to membrane potential. One can only say they are consistent with cell swelling, regardless of what causes the cell swelling. The First line of the discussion still claims that T2 relaxation time and pool size ratio (PSR) can detect responses to membrane potential changes modulated by ionic solutions. However, in the absence of cell swelling controls, this cannot be stated.

For this mechanism to be relevant to measuring neuronal activity directly or explaining techniques such DIANA, one needs to show that the cell swelling changes occur within a millisecond, which has never been reported. If one knows the populations of ECF and pellet, the T2s of the ECF and pellet and the volume change of the cells in the pellet, one can model any expected T2 changes due to neuronal activity. I think one would find that these are minuscule within the context of an action potential, or even bulk action potentials.

Comments on revisions:

The manuscript is well written and my previous methodological concerns have been clarified as well. There are no flaws in the experiments, but the interpretation really depends on simultaneous measurements of cell volume and membrane potential, which have yet to be done.

---

## [Referee Report · Reviewer #2 (Public review)]

Summary:

Min et al. attempt to demonstrate a mechanism whereby magnetic resonance imaging (MRI) can reflect changes in neuronal membrane potentials. They approach this goal by studying how MRI contrast and cellular potentials together respond to treatment of cultured cells with ionic solutions that are known to depolarize or hyperpolarize excitable cells. The authors specifically examine two MRI-based measurements: (A) the transverse (T2) relaxation rate, which reflects microscopic magnetic fields caused by solutes and biological structures; and (B) the fraction or "pool size ratio" (PSR) of water molecules estimated to be bound to macromolecules, using an MRI technique called magnetization transfer (MT) imaging. They see that depolarizing K+ and Ba2+ concentrations lead to T2 increases and PSR decreases that vary approximately linearly with parallel measurements of voltage in a neuroblastoma cell line and that change similarly in a second cell type. They also show that depolarizing potassium concentrations evoke T2 increases in rat brains, and that these changes are reversed when potassium is renormalized. Min et al. argue that their results suggest a basis for noninvasive functional imaging of cellular voltage signals. If this were true, it would help validate a recent paper published by some of the authors (Toi et al., Science 378:160-8, 2022), in which they claimed to be able to detect millisecond-scale neuronal responses by MRI.

Strengths:

The discovery of a mechanism for relating cellular membrane potential to MRI contrast could yield an important means for studying functions of the nervous system. Achieving this has been a longstanding goal in the MRI community, but previous strategies have proven insufficient for neuroscientific or clinical applications. The current paper suggests that one of the simplest and most widely used MRI contrast mechanisms-T2 weighted imaging-may indicate correlates of membrane potential if measured in the absence of the hemodynamic signals that most functional MRI (fMRI) experiments rely on. The authors make their case using quantitative tests that include some controls for ion and cell type-specificity of their in vitro results and reversibility of MRI changes observed in vivo.

Weaknesses:

The major weakness of the paper is that it uses only slow correlational experiments to probe the relationship between MRI contrast and membrane potential. The authors do not examine effects on the subsecond time scale that is of greatest interest, and they do not adequately consider how biophysical factors with only loose relationship to electrophysiological variables could explain their imaging results. Notably, depolarizing ionic solutions that perturb membrane potential can also induce changes in cellular volume and tissue structure that in turn alter MRI contrast properties similarly to the results shown here. For example, a study by Stroman et al. (Magn Reson Med 59:700-6, 2008) reported reversible potassium-dependent T2 increases in neural tissue that correlate closely with light scattering-based indications of cell swelling. Phi Van et al. (Sci Adv 10:eadl2034, 2024) showed that potassium addition to one of the cell lines used here likewise leads to cell size increases and T2 increases. In their revised manuscript, the authors acknowledge that cell swelling might contribute to the MRI signals they report, but they do nothing to probe the contributions or characteristics of such effects. If cell swelling accounted for the author's MRI results, it would likely operate on a time scale far too slow to yield useful indications of membrane potential. Given these considerations and the absence of data demonstrating correspondence of electrophysiological measures with MRI readouts on a fast time scale, the paper fails to provide evidence that membrane potential changes can be meaningfully detected by MRI.

---

## [Author Response]

The following is the authors’ response to the original reviews

**Public Reviews:**

**Reviewer #1 (Public review):**
Summary:This paper examines changes in relaxation time (T1 and T2) and magnetization transfer parameters that occur in a model system and in vivo when cells or tissue are depolarized using an equimolar extracellular solution with different concentrations of the depolarizing ion K^+^. The motivation is to explain T2 changes that have previously been observed by the authors in an in vivo model with neural stimulation (DIANA) and to try to provide a mechanism to explain those changes.Strengths:The authors argue that the use of various concentrations of KCL in the extracellular fluid depolarize or hyperpolarize the cell pellets used and that this change in membrane potential is the driving force for the T2 (and T1-supplementary material) changes observed. In particular, they report an increase in T2 with increasing KCL concentration in the extracellular fluid (ECF) of pellets of SH-SY5Y cells. To offset the increasing osmolarity of the ECF due to the increase in KCL, the NaCL molarity of the ECF is proportionally reduced. The authors measure the intracellular voltage using patch clamp recordings, which is a gold standard. With 80 mM of KCL in the ECF, a change in T2 of the cell pellets of ~10 ms is observed with the intracellular potential recorded as about -6 mv. A very large T1 increase of ~90 ms is reported under the same conditions. The PSR (ratio of hydrogen protons on macromolecules to free water) decreases by about 10% at this 80 mM KCL concentration. Similar results are seen in a Jurkat cell line and similar, but far smaller changes are observed in vivo, for a variety of reasons discussed. As a final control, T1 and T2 values are measured in the various equimolar KCL solutions. As expected, no significant changes in T1 and T2 of the ECF were observed for these concentrations.Weaknesses:[Reviewer 1, Comment 1] While the concepts presented are interesting, and the actual experimental methods seem to be nicely executed, the conclusions are not supported by the data for a number of reasons. This is not to say that the data isn't consistent with the conclusions, but there are other controls not included that would be necessary to draw the conclusion that it is membrane potential that is driving these T1 and T2 changes. Unfortunately for these authors, similar experiments conducted in 2008 (Stroman et al. Magn. Reson. in Med. 59:700-706) found similar results (increased T2 with KCL) but with a different mechanism, that they provide definite proof for. This study was not referenced in the current work.It is well established that cells swell/shrink upon depolarization/hyperpolarization. Cell swelling is accompanied by increased light transmittance in vivo, and this should be true in the pellet system as well. In a beautiful series of experiments, Stroman et al. (2008) showed in perfused brain slices that the cells swell upon equimolar KCL depolarization and the light transmittance increases. The time course of these changes is quite slow, of the order of many minutes, both for the T2-weighted MRI signal and for the light transmittance. Stroman et al. also show that hypoosmotic changes produce the exact same time course as the KCL depolarization changes (and vice versa for the hyperosmotic changes - which cause cell shrinkage). Their conclusion, therefore, was that cell swelling (not membrane potential) was the cause of the T2-weighted changes observed, and that these were relatively slow (on the scale of many minutes).What are the implications for the current study? Well, for one, the authors cannot exclude cell swelling as the mechanism for T2 changes, as they have not measured that. It is however well established that cell swelling occurs during depolarization, so this is not in question. Water in the pelletized cells is in slow/intermediate exchange with the ECF, and the solutions for the two compartment relaxation model for this are well established (see Menon and Allen, Magn. Reson. in Med. 20:214-227, 1991). The T2 relaxation times should be multiexponential (see point (3) further below). The current work cannot exclude cell swelling as the mechanism for T2 changes (it is mentioned in the paper, but not dealt with). Water entering cells dilutes the protein structures, changes rotational correlation times of the proteins in the cell and is known to increase T2. The PSR confirms that this is indeed happening, so the data in this work is completely consistent with the Stroman work and completely consistent with cell swelling associated with depolarization. The authors should have performed light scattering studies to demonstrate the presence or absence of cell swelling. Measuring intracellular potential is not enough to clarify the mechanism.

[Reviewer 1, Response 1] We appreciate the reviewer’s comments. We agree that changes in cell volume due to depolarization and hyperpolarization significantly contribute to the observed changes in T2, PSR, and T1, especially in pelletized cells. For this reason, we already noted in the Discussion section of the original manuscript that cell volume changes influence the observed MR parameter changes, though this study did not present the magnitude of the cell volume changes. In this regard, we thank the reviewer for introducing the work by Stroman et al. (Magn Reson Med 59:700-706, 2008). When discussing the contribution of the cell volume changes to the observed MR parameter changes, we additionally discussed the work of Stroman et al. in the revised manuscript.

In addition, we acknowledge that the title and main conclusion of the original manuscript may be misleading, as we did not separately consider the effect of cell volume changes on MR parameters. To more accurately reflect the scope and results of this study and also take into account the reviewer 2’s suggestion, we adjusted the title to “Responses to membrane potential-modulating ionic solutions measured by magnetic resonance imaging of cultured cells and in vivo rat cortex” and also revised the relevant phrases in the main text.

Finally, when [K^+^]-induced membrane potential changes are involved, there seems to be factors other than cell volume changes that appear to influence T^2^ changes. Our follow-up study shows that there are differences in volume changes for the same T^2^ change in the following two different situations: pure osmotic volume changes versus [K^+^]-induced volume changes. For example, for the same T^2^ change, the volume change for depolarization is greater than the volume change for hypoosmotic conditions. We will present these results in this coming ISMRM 2025 and are also preparing a manuscript to report shortly.

[Reviewer 1, Comment 2] So why does it matter whether the mechanism is cell swelling or membrane potential? The reason is response time. Cell swelling due to depolarization is a slow process, slower than hemodynamic responses that characterize BOLD. In fact, cell swelling under normal homeostatic conditions in vivo is virtually non-existent. Only sustained depolarization events typically associated with non-naturalistic stimuli or brain dysfunction produce cell swelling. Membrane potential changes associated with neural activity, on the other hand, are very fast. In this manuscript, the authors have convincingly shown a signal change that is virtually the same as what was seen in the Stroman publication, but they have not shown that there is a response that can be detected with anything approaching the timescale of an action potential. So one cannot definitely say that the changes observed are due to membrane potential. One can only say they are consistent with cell swelling, regardless of what causes the cell swelling.For this mechanism to be relevant to explaining DIANA, one needs to show that the cell swelling changes occur within a millisecond, which has never been reported. If one knows the populations of ECF and pellet, the T2s of the ECF and pellet and the volume change of the cells in the pellet, one can model any expected T2 changes due to neuronal activity. I think one would find that these are minuscule within the context of an action potential, or even bulk action potential.

[Reviewer 1, Response 2] In the context of cell swelling occurring at rapid response times, if we define cell swelling simply as an “increase in cell volume,” there are several studies reporting transient structural (or volumetric) changes (e.g., ~nm diameter change over ~ms duration) in neuron cells during action potential propagation (Akkin et al., Biophys J 93:1347-1353, 2007; Kim et al., Biophys J 92:3122-3129, 2007; Lee et al., IEEE Trans Biomed Eng 58:3000-3003, 2011; Wnek et al., J Polym Sci Part B: Polym Phys 54:7-14, 2015; Yang et al., ACS Nano 12:4186-4193, 2018). These studies show a good correlation between membrane potential changes and cell volume changes (even if very small) at the cellular level within milliseconds.

As mentioned in the Response 1 above, this study does not address rapid dynamic membrane potential changes on the millisecond scale, which we explicitly mentioned as one of the limitations in the Discussion section of the original manuscript. For this reason, we do not claim in this study that we provide the reader with definitive answers about the mechanisms involved in DIANA. Rather, as a first step toward addressing the mechanism of DIANA, this study confirms that there is a good correlation between changes in membrane potential and measurable MR parameters (e.g., T^2^ and PSR) when using ionic solutions that modulate membrane potential. Identifying MR parameter changes that occur during millisecond-scale membrane potential changes due to rapid neural activation will be addressed in the follow-up study mentioned in the Response 1 above.

There are a few smaller issues that should be addressed.[Reviewer 1, Comment 3] (1) Why were complicated imaging sequences used to measure T1 and T2? On a Bruker system it should be possible to do very simple acquisitions with hard pulses (which will not need dictionaries and such to get quantitative numbers). Of course, this can only be done sample by sample and would take longer, but it avoids a lot of complication to correct the RF pulses used for imaging, which leads me to the 2nd point.

[Reviewer 1, Response 3] We appreciate the reviewer’s suggestion regarding imaging sequences. In fact, we used dictionaries for fitting in vivo T^2^ decay data, not in vitro data. Sample-by-sample nonlocalized acquisition with hard pulses may be applicable for in vitro measurements. However, for in vivo measurements, a slice-selective multi-echo spin-echo sequence was necessary to acquire T^2^ maps within a reasonable scan time. Our choice of imaging sequence was guided by the need to spatially resolve MR signals from specific regions of interest while balancing scan time constraints.

[Reviewer 1, Comment 4] (2) Figure S1 (H) is unlike any exponential T2 decay I have seen in almost 40 years of making T2 measurements. The strange plateau at the beginning and the bump around TE = 25 ms are odd. These could just be noise, but the fitted curve exactly reproduces these features. A monoexponential T2 decay cannot, by definition, produce a fit shaped like this.

[Reviewer 1, Response 4] The T^2^ decay curves in Figure S1(H) indeed display features that deviate from a simple monoexponential decay. In our in vivo experiments, we used a multi-echo spin-echo sequence with slice-selective excitation and refocusing pulses. In such sequences, the echo train is influenced by stimulated echoes and imperfect slice profiles. This phenomenon is inherent to the pulse sequence rather than being artifacts or fitting errors (Hennig, Concepts Magn Reson 3:125-143, 1991; Lebel and Wilman, Magn Reson Med 64:1005-1014, 2010; McPhee and Wilman, Magn Reson Med 77:2057-2065, 2017). Therefore, we fitted the T_2_ decay curve using the technique developed by McPhee and Wilman (2017).

[Reviewer 1, Comment 5] (3) As noted earlier, layered samples produce biexponential T2 decays and monoexponential T1 decays. I don't quite see how this was accounted for in the fitting of the data from the pellet preparations. I realize that these are spatially resolved measurements, but the imaging slice shown seems to be at the boundary of the pellet and the extracellular media and there definitely should be a biexponential water proton decay curve. Only 5 echo times were used, so this is part of the problem, but it does mean that the T2 reported is a population fraction weighted average of the T2 in the two compartments.

[Reviewer 1, Response 5] We understand the reviewer’s concern regarding potential biexponential decay due to the presence of different compartments. In our experiments, we carefully positioned the imaging slice sufficiently remote from the pellet-media interface. This approach ensures that the signal predominantly arises from the cells (and interstitial fluid), excluding the influence of extracellular media above the cell pellet. We described the imaging slice more clearly in the revised manuscript. As mentioned in our Methods section, for in vitro experiments, we repeated a single-echo spin-echo sequence with 50 difference echo times. While Figure 1C illustrates data from five echo times for visual clarity, the full dataset with all 50 echo times was used for fitting. We clarified this point in the revised manuscript to avoid any misunderstanding.

[Reviewer 1, Comment 6] (4) Delta T1 and T2 values are presented for the pellets in wells, but no absolute values are presented for either the pellets or the KCL solutions that I could find.

[Reviewer 1, Response 6] As requested by the reviewer, we included the absolute values in the supplementary information.

**Reviewer #2 (Public review):**
Summary:Min et al. attempt to demonstrate that magnetic resonance imaging (MRI) can detect changes in neuronal membrane potentials. They approach this goal by studying how MRI contrast and cellular potentials together respond to treatment of cultured cells with ionic solutions. The authors specifically study two MRI-based measurements: (A) the transverse (T2) relaxation rate, which reflects microscopic magnetic fields caused by solutes and biological structures; and (B) the fraction or "pool size ratio" (PSR) of water molecules estimated to be bound to macromolecules, using an MRI technique called magnetization transfer (MT) imaging. They see that depolarizing K^+^ and Ba2+ concentrations lead to T2 increases and PSR decreases that vary approximately linearly with voltage in a neuroblastoma cell line and that change similarly in a second cell type. They also show that depolarizing potassium concentrations evoke reversible T2 increases in rat brains and that these changes are reversed when potassium is renormalized. Min et al. argue that this implies that membrane potential changes cause the MRI effects, providing a potential basis for detecting cellular voltages by noninvasive imaging. If this were true, it would help validate a recent paper published by some of the authors (Toi et al., Science 378:160-8, 2022), in which they claimed to be able to detect millisecond-scale neuronal responses by MRI.Strengths:The discovery of a mechanism for relating cellular membrane potential to MRI contrast could yield an important means for studying functions of the nervous system. Achieving this has been a longstanding goal in the MRI community, but previous strategies have proven too weak or insufficiently reproducible for neuroscientific or clinical applications. The current paper suggests remarkably that one of the simplest and most widely used MRI contrast mechanisms-T2 weighted imaging-may indicate membrane potentials if measured in the absence of the hemodynamic signals that most functional MRI (fMRI) experiments rely on. The authors make their case using a diverse set of quantitative tests that include controls for ion and cell type-specificity of their in vitro results and reversibility of MRI changes observed in vivo.Weaknesses:[Reviewer 2, Comment 1] The major weakness of the paper is that it uses correlational data to conclude that there is a causational relationship between membrane potential and MRI contrast. Alternative explanations that could explain the authors' findings are not adequately considered. Most notably, depolarizing ionic solutions can also induce changes in cellular volume and tissue structure that in turn alter MRI contrast properties similarly to the results shown here. For example, a study by Stroman et al. (Magn Reson Med 59:700-6, 2008) reported reversible potassium-dependent T2 increases in neural tissue that correlate closely with light scattering-based indications of cell swelling. Phi Van et al. (Sci Adv 10:eadl2034, 2024) showed that potassium addition to one of the cell lines used here likewise leads to cell size increases and T2 increases. Such effects could in principle account for Min et al.'s results, and indeed it is difficult to see how they would not contribute, but they occur on a time scale far too slow to yield useful indications of membrane potential. The authors' observation that PSR correlates negatively with T2 in their experiments is also consistent with this explanation, given the inverse relationship usually observed (and mechanistically expected) between these two parameters. If the authors could show a tight correspondence between millisecond-scale membrane potential changes and MRI contrast, their argument for a causal connection or a useful correlational relationship between membrane potential and image contrast would be much stronger. As it is, however, the article does not succeed in demonstrating that membrane potential changes can be detected by MRI.

[Reviewer 2, Response 1] We appreciate the reviewer’s comments. We agree that changes in cell volume due to depolarization and hyperpolarization significantly contribute to the observed MR parameter changes. For this reason, we have already noted in the Discussion section of the original manuscript that cell volume changes influence the observed MR parameter changes. In this regard, we thank the reviewer for introducing the work by Stroman et al. (Magn Reson Med 59:700-706, 2008) and Phi Van et al. (Sci Adv 10:eadl2034, 2024). When discussing the contribution of the cell volume changes to the observed MR parameter changes, we additionally discussed both work of Stroman et al. and Phi Van et al. in the revised manuscript.

In addition, this study does not address rapid dynamic membrane potential changes on the millisecond scale, which we explicitly discussed as one of the limitations of this study in the Discussion section of the original manuscript. For this reason, we do not claim in this study that we provide the reader with definitive answers about the mechanisms involved in DIANA. Rather, as a first step toward addressing the mechanism of DIANA, this study confirms that there is a good correlation between changes in membrane potential and measurable MR parameters (although on a slow time scale) when using ionic solutions that modulate membrane potential. Identifying MR parameter changes that occur during millisecond-scale membrane potential changes due to rapid neural activation will be addressed in the follow-up study mentioned in the Response 1 to Reviewer 1’s Comment 1 above.

Together, we acknowledge that the title and main conclusion of the original manuscript may be misleading. To more accurately reflect the scope and results of this study and also consider the reviewer’s suggestion, we adjusted the title to “Responses to membrane potential-modulating ionic solutions measured by magnetic resonance imaging of cultured cells and in vivo rat cortex” and also revised the relevant phrases in the main text.

**Recommendations for the authors:**

**Reviewer #1 (Recommendations for the authors):**
[Reviewer 1, Comment 7] The manuscript is well written. One thing to emphasize early on is that the KCL depolarization is done in an equimolar (or isotonic) manner. I was not clear on this point until I got to the very end of the methods. This is a strength of the paper and should be presented earlier.

[Reviewer 1, Response 7] In response to the reviewer’s suggestion, we have revised the manuscript to present the equimolar characteristic of our experiment earlier.

[Reviewer 1, Comment 8] In terms of experiments, the relaxation time measurements are not well constructed. They should be done with a CPMG sequence with hundreds of echos and properly curve fit. This is entirely possible on a Bruker spectrometer.

[Reviewer 1, Response 8] As noted in our Response to Reviewer 1’s Comment 3, while a CPMG sequence with numerous echoes and straightforward curve fitting can be effective, it is less feasible for in vivo experiments. Our multi-echo spin-echo sequence was a balanced approach between spatial resolution, reasonable scan duration, and the need to localize signals within specific regions of interest.

[Reviewer 1, Comment 9] Measurements of cell swelling should be done to determine the time course of the cell swelling. This could be with NMR (CPMG) or with light scattering. For this mechanism to be relevant to explaining DIANA, one needs to show that the cell swelling changes occur within a millisecond, which has never been reported. If one knows the populations of ECF and pellet, the T2s of the ECF and pellet and the volume change of the cells in the pellet, one can model any expected T2 changes due to neuronal activity.

[Reviewer 1, Response 9] We acknowledge the importance of further research to further strengthened the claims of this study through additional experiments such as cell volume recording. We will do it in future studies.

As noted in our Response 2 to Reviewer 1’s Comment 2, this study does not address rapid membrane potential changes on the millisecond scale, and we acknowledge that establishing the precise timing of cell swelling is crucial for fully understanding the mechanisms of DIANA. Our current work demonstrates that MR parameters (e.g., T^2^ and PSR) correlate strongly with membrane potential-modulating ionic environments, but it does not extend to millisecond-scale neural activation. We recognize the importance of further experiments, such as direct cell volume measurements and plan to incorporate it in future studies to build on the insights gained from the present work.

**Reviewer #2 (Recommendations for the authors):**
Here are a few comments, questions, and suggestions for improvement:[Reviewer 2, Comment 2] I could not find much information about the various incubation times and delays used for the authors' in vitro experiments. For each of the in vitro experiments in particular, how long were cells exposed to the stated ionic condition prior to imaging, and how long did the imaging take? Could this and any other relevant information about the experimental timing please be provided and added to the methods section?

[Reviewer 2, Response 2] We have included the information about the preparation/incubation times in the revised manuscript. For the scan time, it was already stated in the original manuscript: 23 minutes for the single-echo spin-echo sequence and 23 minutes for the inversion-recovery multi-echo spin-echo, for a total of 46 minutes.

[Reviewer 2, Comment 3] In what format were the cells used for patch clamping, and were any controls done to ensure that characteristics of these cells were the same as those pelleted and imaged in the MRI studies? How long were the incubation times with ionic solutions in the patch clamp experiment? This information should likewise be added to the paper.

[Reviewer 2, Response 3] We have clarified in the revised manuscript that SH-SY5Y cells were patch clamp-measured in their adherent state. On the other hand, the cells were dissociated from the culture plate and pelleted, so the experimental environments were not entirely identical. The patch clamp experiments involved a 20–30 minutes incubation period with the ionic solutions. We have included this information in the revised manuscript.

[Reviewer 2, Comment 4] Can the authors provide information about the mean cell size observed under each condition in their in vitro experiments?

[Reviewer 2, Response 4] We did not directly quantify the mean cell size for each in vitro condition in this study, so we do not have corresponding data. However, we acknowledge that this information could provide valuable insights into potential mechanisms underlying the observed MR parameter changes. In future experiments, we plan to include direct cell-size measurements to further elucidate how changes in cell volume or hydration contribute to our MR findings.

[Reviewer 2, Comment 5] The ionic challenges used both in vitro and in vivo could also have affected cell permeability, with corresponding effects that would be detectable in diffusion weighted imaging. Did the authors examine this or obtain any results that could reflect on contributions of permeability properties to the contrast effects they report?

[Reviewer 2, Response 5] We did not perform diffusion-weighted imaging and therefore do not have direct data regarding changes in cell permeability. We agree that incorporating diffusion-weighted measurements could help distinguish whether the MR parameters changes are driven primarily by membrane potential shifts, cell volume changes, or variations in permeability properties. We will consider these approaches in our future studies.

[Reviewer 2, Comment 6] Clearly, a faster stimulation method such as optogenetics, in combination with time-locked MRI readouts of the pelleted cells, would be more effective at demonstrating a useful relationship between cellular neurophysiology and MRI contrast in vitro. Can the authors present data from such an experiment? Is there any information they can present that documents the time course of observed responses in their experiments?

[Reviewer 2, Response 6] In the current study, our methodology did not include time-resolved or dynamic measurements. While it may be possible to obtain indirect information about the temporal dynamics using T^2^-weighted or MT-weighted imaging, such an experiment was beyond the scope of this work. However, we agree that an optogenetic approach with time-locked MRI acquisitions could help directly link cell physiology to MRI contrast, and we will explore this in future studies.

[Reviewer 2, Comment 7] The authors used a drug cocktail to suppress hemodynamic effects in the experiments of Figs. 5-6. What evidence is there that this cocktail successfully suppresses hemodynamic responses and that it also preserves physiological responses to the ionic challenges used in their experiments? Were analogous in vivo results also obtained in the absence of the cocktail?

[Reviewer 2, Response 7] We appreciate the reviewer’s concern regarding pharmacological suppression of hemodynamic effects. Although each component is known to inhibit nitric oxide synthesis, we did not directly measure the degree of hemodynamic suppression in this study. In addition, we cannot definitively confirm that these agents preserved the physiological responses to the ionic challenges. We have clarified these points in the revised manuscript and identified them as limitations of the study.

[Reviewer 2, Comment 8] Why weren't PSR results reported as part of the in vivo experimental results in Fig. 5? Does PSR continue to vary inversely to T2 in these experiments?

[Reviewer 2, Response 8] In our current experimental setup, acquiring the T^2^ map four times required 48 minutes, and extending the scan to include additional quantitative MT measurements for PSR would have significantly prolonged the scanning session. Given that these experiments were conducted on acutely craniotomized rats, maintaining stable physiological conditions for such a long period of time was challenging. Therefore, due to time constraints, we did not perform MT measurements and focused on T_2_ mapping.

[Reviewer 2, Comment 9] The authors have established in vivo optogenetic stimulation paradigms in their laboratory and used them in the Toi et al. DIANA study. Were T2 or PSR changes observed in vivo using standard T2 measurement or T2-weighted imaging methods that do not rely on the DIANA pulse sequence they originally applied?

[Reviewer 2, Response 9] Our current T_2_ mapping experiments utilized a standard multi-echo spin-echo sequence, rather than the DIANA pulse sequence employed in our previous work. In this respect, the T_2_ changes we observed in vivo do not rely on the specialized DIANA methodology.

[Reviewer 2, Comment 10] In the discussion section, the authors state that to their knowledge, theirs "is the first report that changes in membrane potential can be detected through MRI." This cannot be true, as their own Toi et al. Science paper previously claimed this, and a number of the studies cited on p.2 also claimed to detect close correlates of neuroelectric activity. This statement should be amended or revised.

[Reviewer 2, Response 10] We appreciate the reviewer’s comment. We have revised the discussion section of the manuscript to reflect the points raised by the reviewer.

[Reviewer 2, Comment 11] Because the current study does not actually demonstrate that changes in membrane potential can be detected by MRI, the authors should alter the title, abstract, and a number of relevant statements throughout the text to avoid implying that this has been shown. The title, for instance, could be changed to "Responses to depolarizing and hyperpolarizing ionic solutions measured by magnetic resonance imaging of excitable cells and rat brains," or something along these lines.

[Reviewer 2, Response 11] We appreciate the reviewer’s suggestions. We have revised the title, abstract, and relevant statements of the manuscript to clarify that our findings show MR-detectable responses to ionic solutions that are expected to modulate membrane potential, rather than demonstrating direct detection of membrane potential changes by MRI.

[Reviewer 2, Comment 12] The axes in Fig. 3 seem to be mislabeled. I think the horizontal axes are supposed to be membrane potential measured in mV.

[Reviewer 2, Response 12] Thank the reviewer for finding an error. We have corrected the axis labels in Figure 3 to indicate membrane potential (in mV) on the horizontal axis.

[Reviewer 2, Comment 13] Since neither the experiments in Jurkat cells (Fig. 4) nor the in vivo MRI tests (Fig. 5-6) appear to have made in conjunction with membrane potential measurements, it seems like a stretch to refer to these experiments as involving manipulation of membrane potentials per se. Instead, the authors should refer to them as involving administration of stimuli expected to be depolarizing or hyperpolarizing. The "hyperpolarization" and "depolarization" labels of Fig. 4 similarly imply a result that has not actually been shown, and should ideally be changed.

[Reviewer 2, Response 13] To prevent any misleading that membrane potential changes were directly measured in Jurkat cells or in vivo, we have revised the relevant text and figure labels.

[Reviewer 2, Comment 14] The changes in T2 and PSR documented with various K^+^ challenges to Jurkat cells in Fig. 4 seem to follow a step-function-like profile that differs from the results reported in SH-SY5Y cells. Can the authors explain what might have caused this difference?

[Reviewer 2, Response 14] We currently do not have a definitive explanation for why Jurkat cells exhibit a step-function-like response to varying K⁺ levels, whereas SH-SY5Y cells show a linear response to log [K^+^]. Experiments that include direct membrane potential measurements in Jurkat cells would help clarify whether this difference arises from genuinely different patterns of depolarization/hyperpolarization or from other factors. We have revised the revised manuscript to address this point.